# The K/HDEL receptor does not recycle but instead acts as a Golgi-gatekeeper

Jonas C. Alvim [1,3,4], Robert M. Bolt[1,4], Jing An [1], Yasuko Kamisugi [1], Andrew Cuming [1], Fernanda A. L. Silva-Alvim [1,3], Juan O. Concha [2], Luis L. P. daSilva[2], Meiyi Hu[1], Dominique Hirsz[1] & Jurgen Denecke [1] ✉

Accurately measuring the ability of the K/HDEL receptor (ERD2) to retain the ER cargo Amy-HDEL has questioned earlier results on which the popular receptor recycling model is based upon. Here we demonstrate that ERD2 Golgi-retention, rather than fast ER export supports its function. Ligand-induced ERD2 redistribution is only observed when the C-terminus is masked or mutated, compromising the signal that prevents Golgi-to-ER transport of the receptor. Forcing COPI mediated retrograde transport destroys receptor function, but introducing ER-to-Golgi export or *cis*-Golgi retention signals re-activate ERD2 when its endogenous Golgi-retention signal is masked or deleted. We propose that ERD2 remains fixed as a Golgi gatekeeper, capturing K/HDEL proteins when they arrive and releasing them again into a subdomain for retrograde transport back to the ER. An in vivo ligand:receptor ratio far greater than 100 to 1 strongly supports this model, and the underlying mechanism appears to be extremely conserved across kingdoms.

The Golgi apparatus plays a central role in sending and receiving membrane carriers to and from organelles of the secretory pathway. Its evolutionary origin is unknown, and how it maintains its identity despite the constant influx and efflux of transport carriers continues to fascinate and divide the field[1–3]. Membrane-spanning sorting receptors are a particularly interesting class of proteins as they mediate the sorting of cargo but need to be sorted themselves.

A central dogma to describe receptor-mediated transport was first derived from the principle of receptor-mediated endocytosis at the plasma membrane[4]. Another prominent example is the mannose-6-phosphate (M6P) receptor which binds lysosomal proteins in the Golgi, moves in clathrin-coated vesicles to early endosomes, followed by ligand-release and receptor recycling back to the Golgi[5,6]. Continuous recycling[7] permits few receptors to sort many lysosomal proteins, and this was shown to be essential for vacuolar sorting in yeasts[8] and plants[9]. Well-defined receptor mutants with defects in either anterograde or retrograde transport strongly support the recycling model[10,11].

Evidence that a similar recycling principle operates for highly abundant soluble ER residents bearing the KDEL or HDEL retention signal first arose from the detection of post-translational modifications that are Golgi-specific[12,13]. An elegant genetic screen led to the identification of the K/HDEL receptor (ERD2) in yeast[14], followed by the isolation of the human homologue[15]. Ligand-overproduction caused human ERD2 redistribution from the Golgi to the ER[16], and the receptor-recycling principle was generally accepted by the field. Since detection of endogenous ERD2 proved technically difficult[17,18], the field used C-terminal ERD2 fusions for subcellular localisation, to the extent that ligand-induced ERD2 redistribution was used to monitor its activity indirectly[16,19–22], rather than studying ligand-retention directly in function of ERD2 levels.

We have recently established a method that allows quantification of ERD2 activity by monitoring increased retention of HDEL or KDEL proteins[23]. This assay revealed that YFP or RFP fused to the ERD2 C-terminus inactivate the receptor. A fluorescent fusion (YFP-TM-ERD2) in which the ERD2 core remains unobstructed retains the ability

[1]Centre for Plant Sciences, School of Biology, Faculty of Biological Sciences, University of Leeds, Leeds LS2 9JT, UK. [2]Department of Cell and Molecular Biology, Ribeirão Preto Medical School, University of São Paulo, Ribeirão Preto, São Paulo, Brazil. [3]Present address: Laboratory of Plant Physiology and Biophysics, Bower Building, University of Glasgow, Glasgow G12 8QQ, UK. [4]These authors contributed equally: Jonas C. Alvim, Robert M. Bolt. ✉e-mail: j.denecke@leeds.ac.uk

to mediate K/HDEL protein ER retention, but it was only detected at the Golgi apparatus even when ligands were overproduced[23]. Both of these properties were found to be strictly dependent on a native C-terminus harbouring a novel di-leucine motif (LXLPA). We concluded that this signal either mediates extremely fast "frequent flyer" ER export of ERD2 to explain the steady state levels at the Golgi, or that ERD2 may not recycle as currently believed.

Here we have extended our analysis of ERD2 beyond the plant field, showing that its function is highly conserved between plantae, animalia and protozoa. Golgi-residency of ERD2 is conserved between human and plant ERD2 and is dependent on a Golgi-retention mechanism that specifically averts retrograde ERD2 transport to the ER. We provide a systematic set of experiments that strongly reject a "frequent flyer" receptor model, and propose that ERD2 cycles between ligand-bound and ligand-free configurations within the Golgi. The resulting recycling of ligands without receptors explains how very few receptors can handle an enormous surplus of ligands. We discuss the implications for understanding Golgi-identity and how sorting machinery segregates from cargo.

## Results

### YFP-TM-ERD2 can replace endogenous ERD2 in *Nicotiana benthamiana* and *Physcomitrium patens*

Although biochemical validation of the new fluorescent fusion was established using a gain-of-function assay in plant protoplasts[23], we wanted to obtain genetic validation by testing functional complementation in whole organisms. Sequence divergence would allow *Arabidopsis thaliana* ERD2 (AtERD2) to escape the inhibitory effect of a hybrid *N. benthamiana* ERD2ab anti-sense (AS) inhibition strategy.

We first generated stable *Nicotiana benthamiana* transformants co-expressing an *N. benthamiana* ERD2ab AS construct in a T-DNA together with either ST-YFP-HDEL, AtERD2-YFP or YFP-TM-AtERD2 (Fig. 1a). The frequency of callus-formation was very low for ST-YFP-HDEL and AtERD2-YFP constructs. The few shoots obtained revealed the typical ER network for ST-YFP-HDEL (Fig. 1b, first panel). Interestingly, AtERD2-YFP was also mainly found in the ER, although weak mobile punctae were also seen that could be Golgi bodies (Fig. 1b, second panel, white arrow heads). This differs from the dual ER-Golgi localisation of AtERD2-YFP observed previously when expressed from the stronger CaMV35S promoter[19,23]. In both cases fluorescent signals were always very weak, requiring high detector gain settings. Since the ERD2ab AS constructs was expressed from the same T-DNA, it is likely that higher expressing lines did not even reach the shoot stage. Invariantly, shoots from these constructs failed to form roots and all lines were subsequently lost. This illustrates that ERD2 knockdown is lethal[24] and that AtERD2-YFP is non-functional[23].

In sharp contrast, co-expressed YFP-TM-AtERD2 resulted in high-frequency callus formation. 55 shoots were successfully regenerated, all of which rooted and resulted in fertile plants. Leaves from primary transformants showed bright punctate Golgi-structures that were much easier detected (Fig. 1b, third panel). Seeds from the primary transformants germinated normally, and root cortex cells from the resulting seedlings displayed typical mobile Golgi structures (Fig. 1b, last panel). YFP-TM-ERD2 did not reveal any detectable ER network, even at the highest detector gain and contrast. This shows that YFP-TM-ERD2 can compensate for the antisense-mediated ERD2 knockdown in contrast to AtERD2-YFP.

Further validation was also provided in the model bryophyte *Physcomitrium patens* using targeted "knock-in" of YFP-TM-PpERD2, followed by complete deletion of the second PpERD2 gene (Fig. 1c, d). The resulting moss-line showed normal growth revealing YFP-TM-PpERD2 fluorescence in the growing tips and newly formed cell plates (Fig. 1e, panel 1, white stars). Expression was very low and high detector gain settings were needed, therefore also revealing autofluorescence of chloroplasts (Chl.). YFP-TM-PpERD2 was found exclusively in

punctate structures (Fig. 1e, panel 2, white arrow heads) and no structures reminiscent of ER were observed. The results firmly establish that the single YFP-TM-PpERD2 gene expressed under its native promoter can functionally replace both endogenous ERD2 genes in *P. patens*.

### Golgi residency of ERD2 appears to be conserved in eukaryotes

To rule out that ERD2 Golgi residency is a plant-specific feature, ERD2 orthologs from 12 further eukaryotic organisms were first subcloned into the β-glucuronidase (GUS) reference vector and tested in the α-amylase-HDEL (Amy-HDEL) secretion assay in *N. benthamiana* protoplasts[23]. This methodology has been qualitatively validated in situ using fluorescent HDEL-cargo in leaf epidermis tissues[23], but the protoplast system can lead to quantification. While we cannot rule out that trafficking in protoplasts differs from that in intact plant cells or that other cargo molecules may not saturate endogenous receptors, our protoplasts system offers an effective gain-of-function assay where ERD2-activity can be measured in a dose-dependent manner. After equalising the transfection efficiency, verified by obtaining comparable GUS activities from the ERD2 plasmids, the plasmids were co-transfected with plasmids encoding either Amy-HDEL or Amy-KDEL as cargo molecule. Amylase activity was measured in the culture medium and in the cells, followed by calculation of the ratio of extracellular/intracellular enzyme levels, the secretion index.

Figure 2a shows that the vast majority of ERD2 orthologs strongly reduced the secretion index of Amy-HDEL comparable to *Arabidopsis thaliana* ERD2 (AtERD2, second lane). Below a threshold of 50% sequence homology with AtERD2, Amy-HDEL secretion was only weakly reduced (*T. brucei*) or not reduced at all (*K. lactis*, *S. cerevisiae*). *S. cerevisiae* ERD2 (ScERD2 – last lane) consistently induced Amy-HDEL secretion, suggesting a dominant-negative effect on endogenous machinery.

It is interesting that so many ERD2 orthologs were capable of mediating increased capacity for K/HDEL protein retention in plant protoplasts. We were therefore curious to test how the fluorescent plant ERD2 fusions AtERD2-YFP and YFP-TM-AtERD2 would localise if expressed in human cells. Figure 2b shows the fluorescent pattern of both constructs when co-transfected with RFP-KDEL as ER marker, and using endogenous GM130 as cis-Golgi marker present in all cells. AtERD2-YFP is mostly found in the ER, although partial overlap with the cis-Golgi marker was also visible and more pronounced when expressed at higher levels. At low levels of expression, AtERD2-YFP was mostly found in the ER. In sharp contrast, YFP-TM-AtERD2 was undetectable in the ER in all conditions, and co-localised with the cis-Golgi marker. This corresponds extremely well with findings in plant cells[23] and suggests the mechanism of ERD2 Golgi localisation could be conserved between plants and mammals. Further experiments reveal that YFP-TM-AtERD2 is most likely concentrated in the cis-Golgi because it does not co-localise well with the trans-Golgi marker TGN46 (Fig. 2c). The two fluorescent fusion variants were also constructed with human ERD2 (HsERD2) and showed the same localisation patterns in HeLa cells (Fig. 2d).

### Golgi residency of ERD2 is strictly linked to functionality in eukaryotes

When expressed in plant protoplasts, HsERD2 mediates a very similar ER retention activity as AtERD2 (Fig. 2a). This is further illustrated by a dose–response assay that compares the plant and human receptors (Supplementary Fig. 3a). More importantly, HsERD2 shares the same sensitivity to C-terminal fusions as seen for plant ERD2 (Fig. 3a). YFP-TM-HsERD2 was found exclusively in the Golgi (upper panel) whilst the C-terminal fusion (HsERD2-YFP) displayed a dual ER-Golgi localisation (Fig. 3a, lower panel). The former fusion promoted strong Amy-HDEL retention whilst the latter did not (Fig. 3b, second and third lane). This shows that our previous results on plant ERD2 fusions[23] can be

reproduced with human ERD2. Together with the results from Fig. 2b–d, the findings show that the requirement for ERD2 Golgi-residency to be biologically active is not a plant-specific property.

An earlier study of the human ERD2 C-terminus proposed that Serine209 controls ARF-GAP recruitment via PKA phosphorylation[21]. In our activity assay, neither the inactive (S209A) nor the phosphomimetic (S209D) mutation affected human ERD2 function in our assay

(Fig. 3b). Moreover, this residue is not conserved between plant and human ERD2 (Fig. 3c).

More recently, a "cluster of lysines" near the ERD2 C-terminus was proposed to form an inducible retrieval motif for coat protein complex I (COPI) Golgi to ER transport[22]. Here we show that mutating each lysine individually (K206, K207) or combined as a double mutation does not alter the biological activity of human ERD2 (Fig. 3b). This is

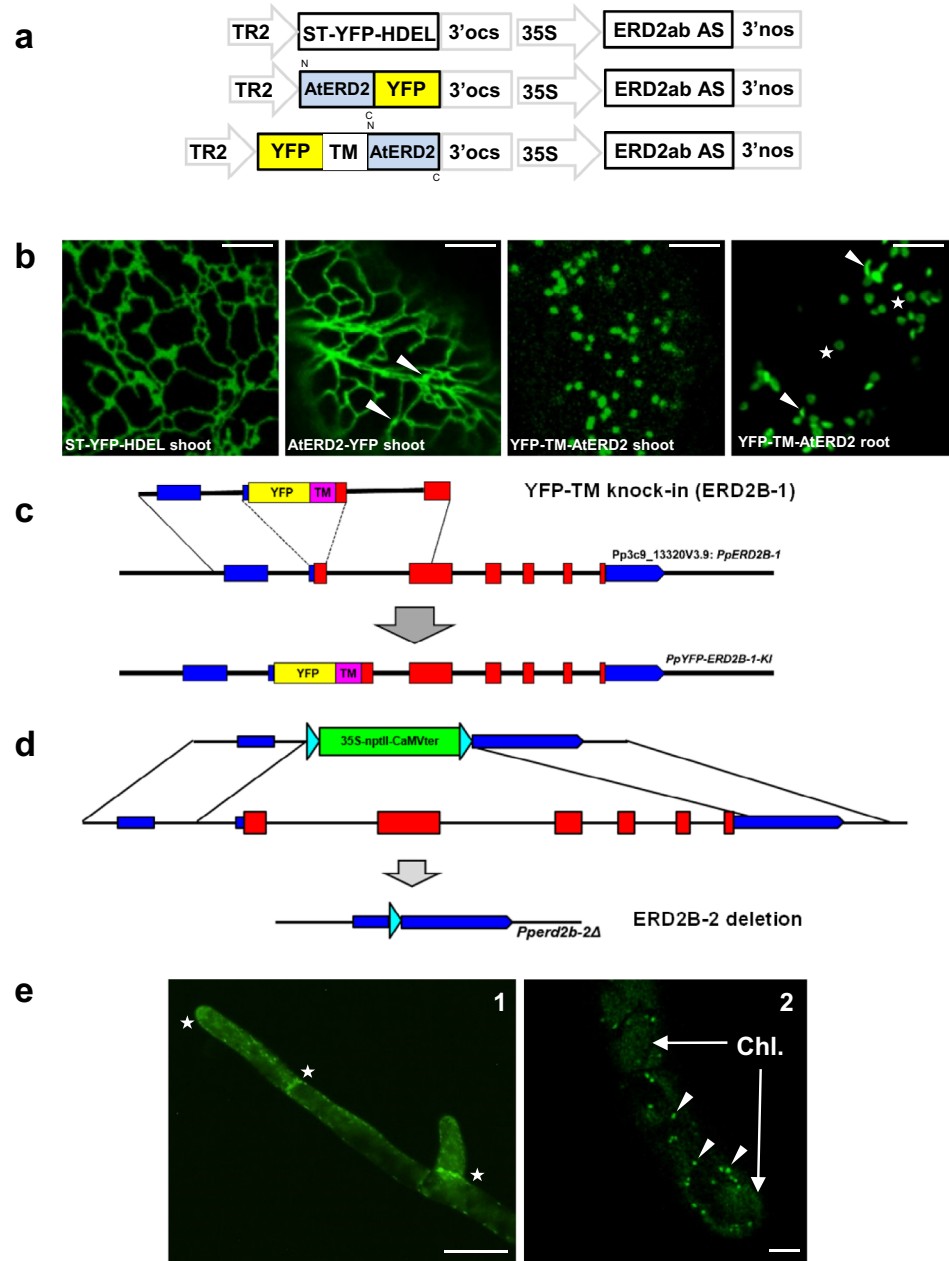

**Fig. 1 | Genetic validation of YFP-TM-ERD2 by stable transformation in *Nicotiana benthamiana* and *Physcomitrium patens*. a** Schematic of *N. benthamiana* ERD2ab antisense (AS) construct driven by the strong constitutive CaMV35S promoter (35S), combined with YFP constructs expressed under the weak TR2 promoter on the same T-DNA. **b** Confocal laser scanning microscopy (CLSM) of stably transformed *N. benthamiana* tissues, expressing all three fusions in regenerating shoots in tissue culture. Only YFP-TM-ERD2 led to fertile plants allowing us to image this fusion in root cortex cells from next-generation seedlings. Notice that ST-YFP-HDEL labels the ER, ERD2-YFP labels the ER and weak Golgi bodies, while YFP-TM-ERD2 only labels Golgi bodies. In roots, Golgi-stacks are either viewed from the side (arrow heads) or from top/bottom (stars), giving rise to the typical donut shapes.

Even with high detector gain, YFP-TM-ERD2 cannot be detected in the ER. Size marker 10 μm. **c** Schematic of YFP-TM targeted gene knock-in onto *Pp*ERD2B-1 (Pp3c9_13230V3.9), leading to expression of a YFP-TM-ERD2 derivative under the transcriptional control of the native promoter in *P. patens*. **d** Schematic of *Pp*ERDB2-2 (Pp3c15_12830) knockout by complete deletion of the second ERD2 gene. **e** YFP-TM-PpERD2 expression under its native promoter in *P. patens*. (e1) Notice stronger expression near growing tips and newly formed cell plates (white stars). Size marker 50 μm. (e2) At high magnification, distinguish punctate structures (white arrow heads) from weak autofluorescence of chloroplasts (Chl.). Size marker 10 μm.

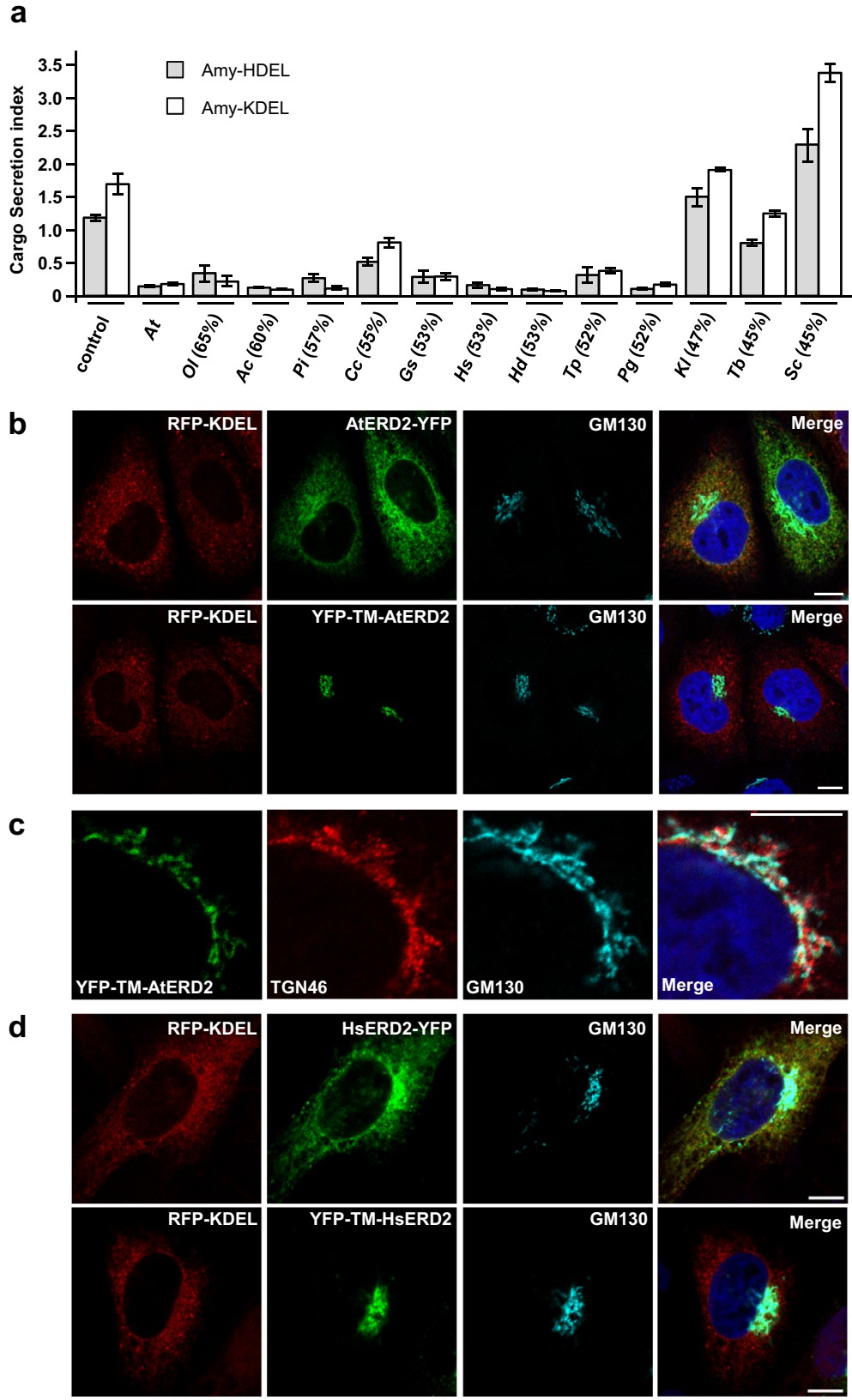

particularly obvious from a dose–response assay (Supplementary Fig. 3b). In contrast, the conserved leucines at position −3 and −5 from the C-terminus (Fig. 3c) are essential for human ERD2 activity as the double mutant has lost all activity (Fig. 3b, LLGG, last lane). Figure 3d shows that all human ERD2 mutants with wild type Amy-HDEL retention activity are strictly Golgi-localised whilst additional labelling of the

ER network was only seen when the two conserved leucines were mutated (LLGG).

To explore if the ERD2-C-terminus can tolerate small epitopes as for instance the c-myc epitope[16], we tested 3 different tags on either plant or human ERD2 for activity and localisation (Fig. 3e). FLAG- and c-myc-tagged ERD2 showed strongly reduced biological activity

**Fig. 2 | ERD2 function and Golgi residency is conserved amongst eukaryotes.**
**a** Retention assay using protoplasts showing the secretion index (ratio extra/intracellular Amy-HDEL activity) with cargo alone (either Amy-HDEL or Amy-KDEL) or with co-expressed *A. thaliana* ERD2b (At) and 12 further ERD2 orthologs from the eukaryotes *Ostreococcus lucimarinus* (Oi), *Acanthamoeba castellanii* (Ac), *Phytophthora infestans* (Pi), *Chondrus crispus* (Cc), *Galdieria sulphuraria* (Gs), *Homo sapiens* (HsERD2), *Hypsibius dujardini* (Hd), *Thalassiosira pseudonana* (Tp), *Puccinia graminis* (Pg), *Kluyveromyces lactis* (KlERD2), *Trypanosoma brucei* (Tb) and *Saccharomyces cerevisiae* (Sc). Transfection efficiencies were normalised by the internal marker GUS established at 5 standard OD units as described in materials and methods. Percentages in brackets refer to the sequence identity with AtERD2b. Error bars are standard errors. Source data are provided as a Source Data file.

**b** CLSM analysis of two separate Arabidopsis thaliana ERD2b fluorescent fusions (YFP-TM-AtERD2 and AtERD2-YFP, shown in green), each co-expressed with the ER marker RFP-KDEL (shown in red) in HeLa cells. The endogenous cis-Golgi marker GM130 was detected via immunocytochemistry (shown in light blue) and the nucleoplasm is stained with DAPI (dark blue). Notice that in contrast to the C-terminal AtERD2-YFP fusion, YFP-TM-AtERD2 is not detected in the ER and shows the best co-localisation with GM130. The size marker bar is 10 microns. **c** CLSM analysis at higher magnification to compare YFP-TM-AtERD2 with two different Golgi markers. Notice that the trans-Golgi marker TGN46 is clearly distinct from the ERD2-fusion and GM130. Size marker 10 microns. **d** As in (**b**), but fluorescent fusions contain human ERD2 (HsERD2). Size marker 10 microns.

compared to untagged ERD2 (Fig. 3f) and were localised to both the ER and the Golgi (Fig. 3g). A dose–response assay illustrates how strongly those two tags affected the functionality of ERD2 (Supplementary Fig. 3c) although weak residual activity may suffice for yeast complementation[14]. Interestingly, fusion of the HA tag (YPYDVPDYA) had only a minor effect on either plant or human ERD2 activity (Fig. 3f) and resulted in predominant Golgi localisation with almost none detected in the ER, unless very high detector gain settings are used (Fig. 3g). This exception highlights the temperamental nature of C-terminal modifications, but also shows the strict correlation between Golgi residency and strong biological activity.

Given the combined results, we propose that conclusions based on experiments with C-terminal ERD2 fusions, in particular those with large fluorescent proteins such as ERD2-YFP, should be re-considered. Due to the consistent outcomes of experiments with both plant and human ERD2 variants, as well as similar localisation data in both plant and mammalian cell models, all further experiments were carried out with AtERD2 in our plant model system, as it also offers highly quantitative cargo secretion assays.

## The C-terminal di-leucine motif of ERD2 prevents its recycling to the ER
The "frequent flyer" model predicts that the di-leucine motif is a fast ER to Golgi export signal, and that mutating the signal would slow down ER to Golgi transport and cause partial ER retention of ERD2. Fluorescence recovery after photobleaching (FRAP) experiments revealed that YFP-TM-ERD2 arrival at the Golgi is not slowed down by mutating the signal (Fig. 4a). An ER export function of the di-leucine motif can thus be ruled out. Instead the data suggest that the di-leucine motif is in fact a Golgi-retention signal that specifically prevents Golgi to ER retrograde transport. The mutant can undergo retrograde Golgi to ER transport and generates a visible ER resident pool of ERD2 (Supplementary Fig. 4c) greater than what is supplied by de novo synthesis which is below the detection limit. This explains how the mutant recovers faster at the Golgi (Fig. 4a, Supplementary Fig. 4c). It should also be noted that the wild type ERD2 fusion does not exhibit unusually fast "frequent-flyer" ER export to start with, because it recovers at the same rate as the Golgi marker ST-YFP (Supplementary Fig. 4a, b).

To show that ligand-induced ERD2 redistribution to the ER[20,22,25] could be caused by compromised Golgi-retention, we co-expressed secreted Amy or retained Amy-HDEL together with either functional or non-functional ERD2-fusions (Fig. 4b). YFP-TM-ERD2 shows no HDEL-mediated redistribution (Fig. 4c). Deletion of the last 5 amino acids LQLPA (YFP-TM-ERD2ΔC5) results in partial ER localisation which is strongly exacerbated upon HDEL-overdose (Fig. 4d). The same HDEL ligand-induced redistribution from Golgi to ER can be seen with ERD2-YFP (Fig. 4e). Supplementary Fig. 5 illustrates how the Golgi signal of ERD2-YFP is drastically reduced upon HDEL-protein co-expression (compare panel a with panel b). Interestingly, low expression of ERD2-YFP alone also favours an ER-localisation (Fig. 4f), similar to the transgenic lines expressing ERD2-YFP at low levels (Fig. 1b). We also

noticed in HeLa cells that the C-terminal fusion is mostly ER localised when expressed at low levels (Supplementary Fig. 5c).

The combined results show that ligand-mediated ER redistribution of ERD2 is an artefact that is only observed when the di-leucine motif is mutated or masked. Interestingly, ERD2 leakage to post-Golgi organelles remains undetectable in all conditions and cellular models tested, suggesting that an additional mechanism prevents post-Golgi transport of ERD2.

## The Golgi-retention motif is required to inhibit COPI-mediated receptor recycling
It has been reported that ERD2 can cause mixing of Golgi and ER membranes[26] by recruiting ARF1-GAP to the Golgi apparatus[27]. We therefore tested the influence of wild type and mutant ERD2 on the localisation of the GTPase ARF1. An ARF1-RFP fusion typically co-localises with the Golgi-marker ST-YFP and additional post-Golgi structures (Fig. 5a, white arrow heads), but when co-expressed with wild type ERD2, ST-YFP redistributes to an ER-like super-compartment and ARF1 redistributes to the cytosol. This is accompanied by a reduction in constitutive secretion. ERD2 inhibits Amy-HDEL secretion at low expression levels without affecting constitutive Amy secretion[23] but when very high levels of ERD2 plasmids are co-transfected, Amy secretion is also inhibited (Fig. 5b). All three observations are reminiscent of Brefeldin A treatment[19]. The deletion mutant ERD2ΔC5 has no effect on the localisation of ST-YFP and ARF1-RFP when compared to the control (Fig. 5a, upper and lower panel). Likewise, ERD2ΔC5 affects neither Amy-HDEL retention nor Amy secretion (Fig. 5b). This shows that a functional Golgi-resident ERD2 is required to cause these Brefeldin A-like effects.

The results appear to be in conflict with an earlier report of protein-protein interactions between ERD2 and ARF1[25], but these were based on the use of inactive C-terminal ERD2 fusions. Our current findings strongly suggest that increasing levels of ERD2 compromise COPI-mediated transport, and that the Golgi-retention signal is required for this. We postulate that at low native expression levels, ERD2 only prevents its own entry into COPI carriers, while higher expression leads to a complete collapse of the ER-Golgi system.

We also analysed the effect of Brefeldin A on the ERD2 Golgi localisation. Compared to the Golgi-marker ST-YFP, RFP-TM-ERD2 persisted slightly longer at the Golgi but generally redistributed readily to the ER following prolonged Brefeldin A treatment (Fig. 5c).

## COPII/COPI-mediated recycling of ERD2 is incompatible with its biological function
To test COPI-dependence more directly, we created an ERD2 hybrid (Fig. 5d) by replacing the 9 most C-terminal amino acids with those of a p24 family member (p24δ5, AT1G21900). This results in an ERD2-hybrid (ERD2-p24) which has the same overall size as wild type ERD2 but instead of harbouring a Golgi-retention motif, it now contains a well-characterised di-hydrophobic sequence (YF) for coat protein complex II (COPII)-mediated ER export and a canonical di-lysine motif (KKXX) for COPI-mediated recycling[28,29].

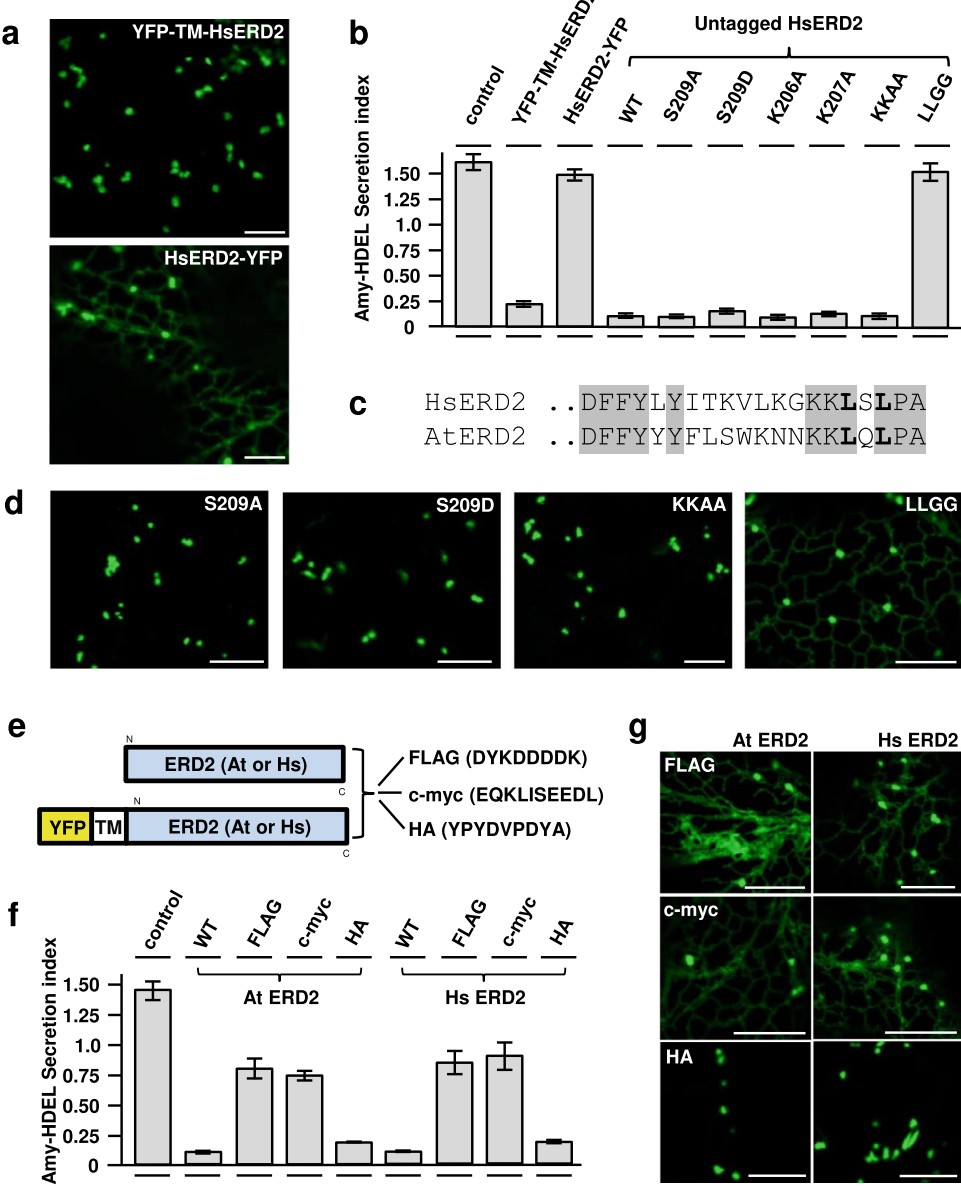

**Fig. 3 | Analysis of C-terminal residues and sensitivity to C-terminal epitope tagging in Hs and At ERD2. a** CLSM analysis of two human ERD2 fusions (YFP-TM-HsERD2 and HsERD2-YFP constructed as described (Silva-Alvim et al., 2018) imaged in tobacco leaf epidermis cells. Notice that only the C-terminal YFP fusion causes partial ER localisation. Size marker 10 microns. **b** Amy-HDEL retention assays as in Fig. 2a but either comparing the two HsERD2 fluorescent variants from panel (**a**), or a comparison of untagged HsERD2 (WT) with the point–mutations in the HsERD2 C-terminus indicated above each lane. Notice that the C-terminal YFP fusion has completely lost biological activity. Notice also that only the LLGG mutant has lost biological activity when untagged HsERD2 is analysed. Error bars are standard deviations from 3 biological replicas. A full dose response for KKAA is provided in Supplementary Fig. 3. **c** C-terminal amino acid sequences of human (KDELR2) and *A. thaliana* ERD2B. Conserved residues are highlighted grey and the conserved di-leucine motif is highlighted bold. **d** CLSM analysis of selected mutants from (**b**) but in the YFP-TM-HsERD2 configuration. Silent mutations in panel (**b**) retain the Golgi localisation, while the inactive LLGG mutant displays partial ER localisation. **e** Schematic of C-terminal fusions to At and Hs ERD2 for functional assays (upper) and the fluorescent derivative for CLSM analysis (lower schematic). **f** Secretion index of Amy-HDEL, co-expressed with either wild type human or plant ERD2 compared to the three different C-terminal modifications (FLAG, c-myc, HA) in each case. Constant levels of ERD2 encoding plasmids were co-transfected (yielding 5 standard OD units). In both instances, the addition of a FLAG or c-myc tag strongly reduced function, whilst most of the activity was maintained for each ortholog after adding the HA tag. Error bars are standard deviations from 3 biological replicas. Source data are provided as a Source data file. **g** Localisation of human and plant ERD2 fluorescent fusions with C-terminal FLAG, c-myc and HA tags. Notice that FLAG and c-myc additions cause an ER-Golgi localisation, whilst the addition of an HA tag does not affect the Golgi localisation of YFP-TM-ERD2 for both orthologs.

When expressed in protoplasts, ERD2-p24 showed no biological activity on HDEL cargo (Fig. 5e). However, mutating the two lysines of ERD2-p24 into serines (KKSS), and thus eliminating the signal for COPI-mediated recycling induced receptor activity again (last 3 lanes). In order to relate these findings to the subcellular localisation of the ERD2-variants, we introduced the same changes in the RFP-TM-ERD2 backbone (Fig. 5f). CLSM analysis revealed complete ER retention of

ERD2-p24, with no detection in the Golgi bodies, illustrating the dominant nature of the COPI sorting signal. Again, destroying the COPI signal by replacing the two lysines with serines (KKSS) re-establishes Golgi-residency (Fig. 5f). The KKSS mutant is also sensitive to HDEL ligand-induced redistribution to the ER, indicating that its Golgi residency is not caused by retention but faster ER export instead (Fig. 5f, bottom panel).

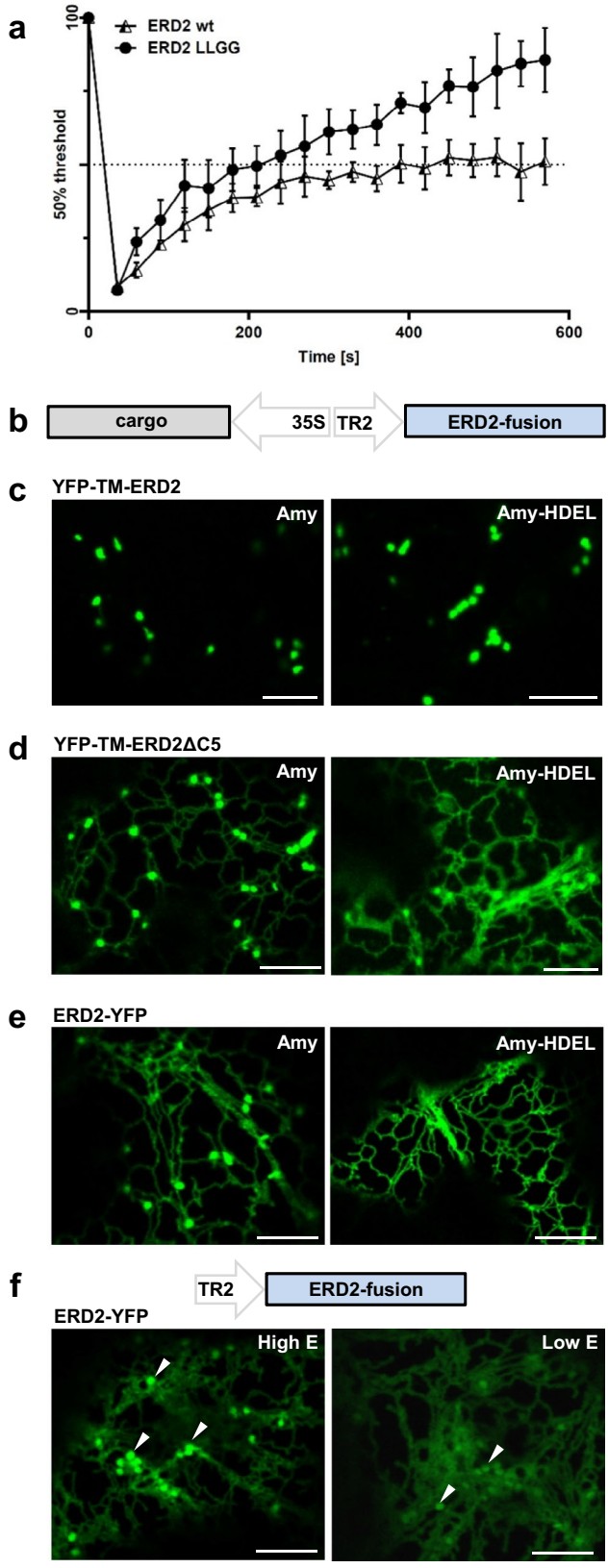

**Fig. 4 | FRAP and redistribution assays identify a Golgi-retention signal at the ERD2 C-terminus. a** Fluorescence recovery after photobleaching (FRAP) comparing wild type ERD2 and the LLGG mutant. ERD2 recovery to 50% (400 s) took almost twice the time of the LLGG mutant (240 s). LLGG mutant recovery reached 85%, whereas wild type ERD2 remained at around 50%. Error bars are standard deviations from at least 6 biological replicas. Source data are provided as a Source data file. **b** Schematic of dual expression T-DNA constructs used to co-express fluorescent ERD2 fusions with either secreted Amy or the ERD2-ligand Amy-HDEL. **c** Golgi localisation of YFP-TM-ERD2 co-expressed with Amy and Amy-HDEL. Distribution remains unchanged for both cargo. Size bars 10 microns. **d** Dual ER-Golgi localisation of YFP-TM-ERD2ΔC5 co-expressed with Amy and a more prominent ER localisation when co-expressed with Amy-HDEL. Size bars 10 microns. **e** Dual ER-Golgi localisation of ERD2-YFP co-expressed with Amy. The re-distribution of ERD2-YFP to the ER by co-expressed Amy-HDEL is even more drastic compared to that of the deletion mutant in panel (**d**). Size bars 10 microns. **f** Schematic including the T-DNA construct used to study ERD2-YFP localisation, which is variable depending on expression levels. Golgi bodies (arrows) are labelled by ERD2-YFP more visibly during high cellular expression, whereas punctae are much fainter relative to the ER fluorescence at low expression levels (imaged at higher detector gain). Size bars 10 microns.

## An alternative Golgi-retention signal can re-activate ERD2-YFP

To test if Golgi-retention per-se promotes ERD2 function, we selected the newly identified N-terminal cytosolic Golgi retention motif (LPYS) of *Arabidopsis thaliana* α-mannosidase I (MNS3) shown to mediate cis-Golgi localisation[30]. We first show that YFP-TM-ERD2 co-localises better with MNS3-RFP than with ST-RFP (Fig. 6a, b). This suggests that ERD2 is mainly retained at the cis-Golgi and corresponds well with the co-localisation with the cis-Golgi marker GM130 in HeLa cells (Fig. 2c). In a second step, we supplemented inactive ERD2-YFP with the same N-terminal TM domain as in previous constructs (TM-ERD2-YFP) and then introduced the MNS3 N-terminus harbouring the LPYS signal (LPYS-TM-ERD2-YFP) to see if it can compensate for the masked C-terminal di-leucine motif.

The Amy-HDEL cargo sorting assay (Fig. 6c) confirms that TM-ERD2 has almost wild-type activity[23]. In contrast, TM-ERD2-YFP has lost this activity due to the devastating effect of C-terminally fused YFP. However, LPYS-TM-ERD2-YFP was partially re-activated despite the masked C-terminus. TM-ERD2-YFP remains partially ER-localised while LPYS-TM-ERD2-YFP is exclusively detected in the Golgi bodies (Fig. 6d).

These results provide experimental evidence directly arguing against the receptor recycling model.

## ERD2 mediates extra-stoichiometric retention of HDEL proteins

Having observed extremely low ERD2 expression levels under a native promoter (Fig. 1e), we also realised that dose–response assays invariantly used much lower plasmid concentrations for the wild-type ERD2 plasmid relative to the Amy-HDEL cargo plasmid. We thus wanted to determine the in vivo ratio between introduced ERD2 molecules and redistributed ligands, a question that has never been addressed quantitatively in any system.

HA-tagged ERD2 had the highest biological activity of all tagged ERD2 variants (Fig. 3f), therefore we expressed this receptor variant and Amy-HDEL from two separate GUS reference vectors. After adjusting the plasmid concentrations to achieve comparable GUS levels when transfected individually, we co-transfected the two plasmids and incubated the protoplasts for 8 h of continuous metabolic $^{35}$S labelling. Quantitative immunoprecipitation with excess antibodies was used to establish relative numbers of ectopically expressed ERD2 versus Amy-HDEL proteins in vivo.

Cell extracts were either immunoprecipitated with anti-HA antibodies to quantify relative ERD2-HA numbers, or with anti-amylase antibodies to quantify relative ligand numbers. Ligands were also immunoprecipitated from an amount of culture medium that

The results show that canonical COPI transport is clearly incompatible with ERD2 function. However, the proposed COPII ER export signal in the p24 C-terminus[29] seems to compensate for the lack of a dedicated Golgi-retention signal. Regardless of the mechanism, the results echo the previously observed correlation between Golgi residency and activity of ERD2.

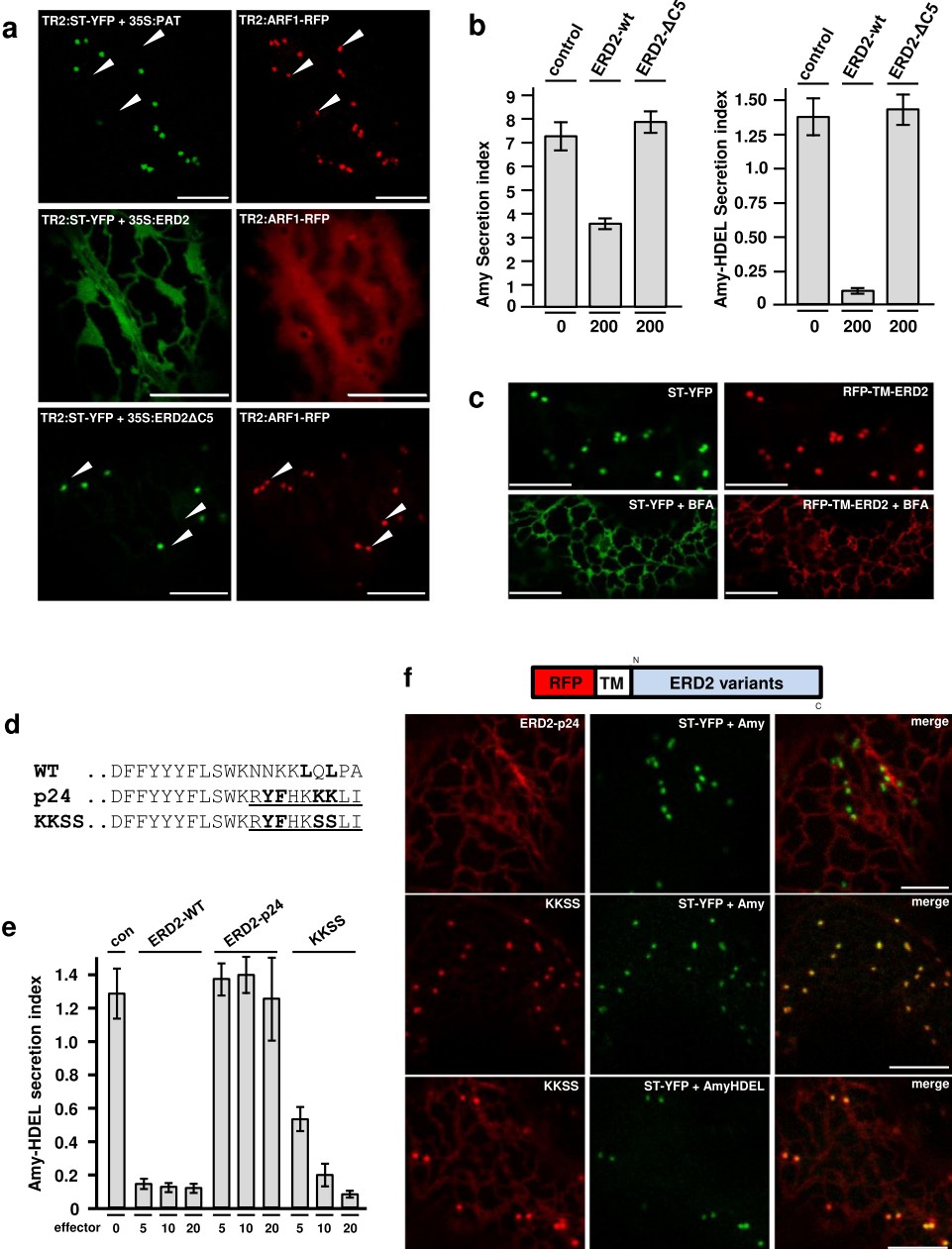

**Fig. 5 | A canonical COPI transport motif (KKXX) causes ER Localisation and abolishes Amy-HDEL retention of ERD2. a** C-terminal amino acid sequences of ERD2 wild type (WT) and two variants in which the last 9 amino acids of ERD2 is replaced by the corresponding region of p24 (underlined). The proposed COPII ER export signal of p24 (Contreras et al., 2004b) is in bold, as is the dileucine motif in the WT sequence, the relevant lysines of the canonical COPI transport motif in the p24 variant and finally the mutant serines in the KKSS variant. Size bars 10 microns. **b** Dose–response assay measuring the influence of co-transfected C-terminal ERD2 variants (given in standard GUS OD units below each lane) on Amy-HDEL secretion. ERD2-WT mediates strong cell retention whilst the p24 fusion shows no retention activity. The KKSS mutant of the p24 fusion restores the retention activity at the highest dose. Error bars are standard deviations from at least 4 biological replicas. **c** The effect of Brefeldin A on the transport of RFP-TM-ERD2 compared to ST-YFP. Notice that both fusions have re-distributed to the ER after 3 h of Brefeldin A treatment. Size bars 10 microns. **d** Sequence of the ERD2 C-terminus, the p24 fusion and the KKSS mutant thereof. **e** Amy-HDEL retention activity of constructs presented in (**d**). Fusing the p24 C-terminus renders ERD2 completely inactive, yet mutating the KKXX motif restores the bulk of biological activity. However, KKSS cannot meet the activity of the wild type ERD2 at lower doses. Error bars are standard deviations from at least 4 biological replicas. Source data are provided as a Source data file. **f** Localisation of p24 and KKSS hybrids incorporated into fluorescent ERD2 fusion proteins. The p24 C-terminus mediates complete ER localisation of the resulting ERD2 fusion whilst the KKSS mutant shows high steady-state levels at the Golgi. Size bars 10 microns.

corresponded to the amount of cell extract. Despite equalised plasmid transfection, ERD2-HA produces a weak signal in the cells which is dwarfed by the high levels of cellular and secreted Amy-HDEL (Fig. 7a). Nevertheless, ERD2-HA co-expression reduced secreted Amy-HDEL levels, accompanied by an increase in the cells. Since exactly two-thirds of either cysteine or methionine are found in ERD2 when compared to Amy-HDEL (Fig. 7b), the relative number of molecules could be determined by phosphor imaging and multiplying ERD2 values by a factor 1.5. Amy-HDEL radioactivity is extremely high compared to that of ERD2-HA (1184 units), increasing from 23,098 to 29,512 units in the cells due to co-expressed ERD2-HA. Correcting for the number of cysteine and methionine residues, the introduced ERD2-HA is the equivalent of 1777 units, approximately 4.5-fold lower than the increase in cellular Amy-HDEL molecules (8026 units). This shows that

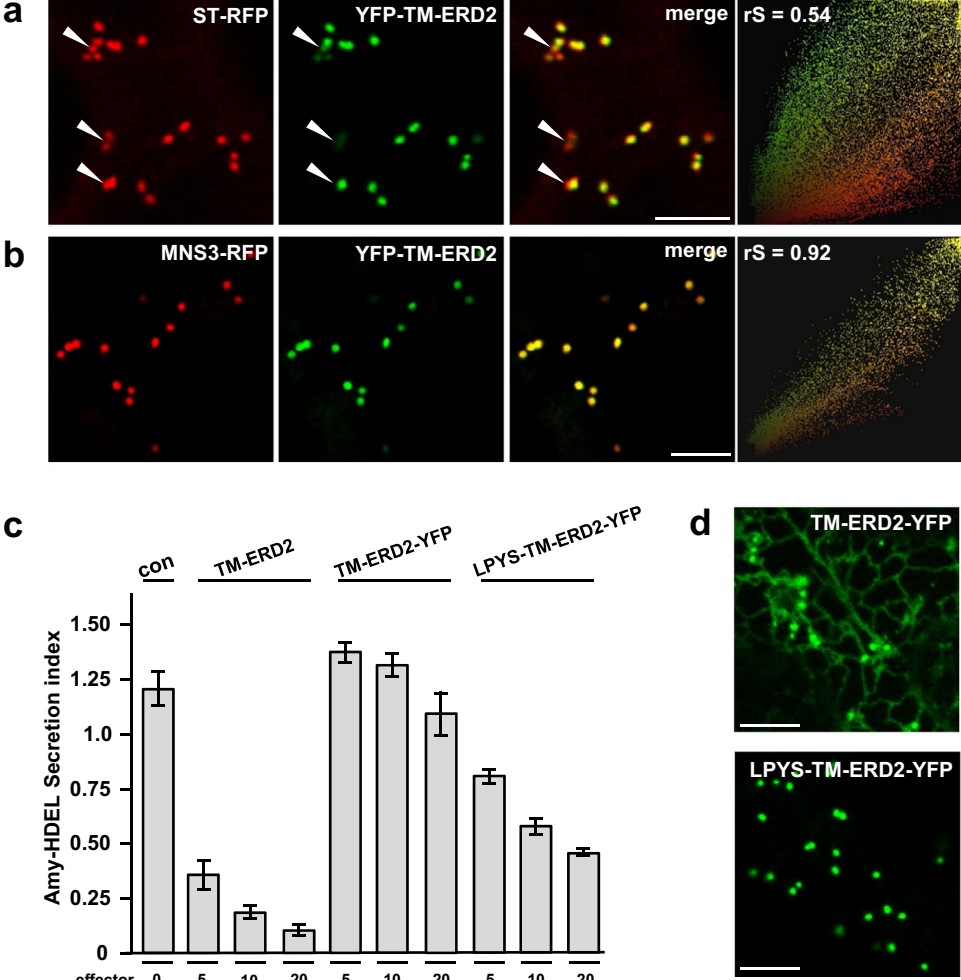

**Fig. 6 | Reactivation of ERD2-YFP with a novel cis-Golgi retention motif.**
**a** Comparative Golgi distribution between ST-RFP and YFP-TM-ERD2. Although both labelling punctae, an rS value of 0.54 highlights a cis-trans Golgi segregation of YFP-TM-ERD2 and ST-RFP, respectively. This is most visible in regions with mostly red fluorescence (white arrow heads), also apparent from two populations in the scatterplot. Size bars 10 microns. **b** When compared with MNS3-RFP instead, the shared cis-Golgi localisation of YFP-TM-ERD2 is clearly demonstrated by the much higher rS value of 0.92 and a single population in the scatterplot. Size bars 10 microns. **c** Retention of Amy-HDEL by ERD2 fusion variants at increasing

concentrations. As previously published, TM-ERD2 effectively retains Amy-HDEL at low and high concentrations. Meanwhile, the C-terminal addition of YFP completely abolishes retention, regardless of increasing concentration. However, the N-terminal addition of the LPYS Golgi retention motif does allow significant activity to return with increasing concentration. Error bars are standard deviations from at least 2 biological replicas. Source data are provided as a Source data file.
**d** N-terminal addition of LPYS causes redistribution of TM-ERD2-YFP exclusively to the Golgi, in agreement with reactivation in secretion assays. Size bars 10 microns.

there are approximately 4.5 additional Amy-HDEL molecules recovered in the cells for each ERD2 molecule introduced (Fig. 7c).

A dose–response experiment with the longer standard 24 h incubation was carried out using the same transfection conditions for Amy-HDEL and ERD2-HA as in Fig. 7a to establish a baseline (Fig. 7d, first two lanes). Measurement of GUS levels confirmed that both plasmids were transfected at comparable rates, and that the two plasmids together produced approximately twice the GUS level as expected (Supplementary Fig. 6a, first 3 lanes). Further transfections were included in which the Amy-HDEL plasmid was kept constant but the ERD2 plasmid was progressively diluted up to 100-fold (Fig. 7d, all further lanes). Maximal Amy-HDEL retention was sustained up to 20-fold dilution of the ERD2-HA plasmid, which indicates a 1 to 90 ratio when considering the 1 to 4.5 ratio at the start (Fig. 7c). 50-fold and 100-fold dilutions showed only a weak reduction in Amy-HDEL retention suggesting that the ratio is actually higher. We cannot detect in vivo labelled ERD2 under these conditions, but the internal marker GUS illustrates the quantitative nature of the dose–response assay in plant protoplasts (Supplementary Fig. 6a, b).

From these data we can ascertain that one introduced ERD2 molecule can prevent the secretion of at least 100 Amy-HDEL proteins. The true extra-stoichiometric retention capacity is likely to be much higher, because plasmid co-transfection is never 100% complete and ERD2-HA is slightly less active than wild-type ERD2. Finally, a protein can only be secreted once, but retention in the ER requires endless recycling, as discussed below.

## Discussion

Results presented here further establish the predominant Golgi residency of biologically active ERD2 fusions[23] and strongly support an emerging relationship between Golgi-retention and receptor-function. We started with genetic complementation assays to demonstrate that YFP-TM-ERD2 is biologically active while ERD2-YFP is not (Fig. 1). Next we demonstrate that ERD2 function is highly conserved in eukaryotes, to the extent that human ERD2 can sort HDEL-and KDEL proteins in plant cells (Fig. 2). Plant and human ERD2 are both localised to the Golgi apparatus in plant as well as human cells, as long as the C-terminus is not masked by fusing proteins or peptides (Figs. 2, 3).

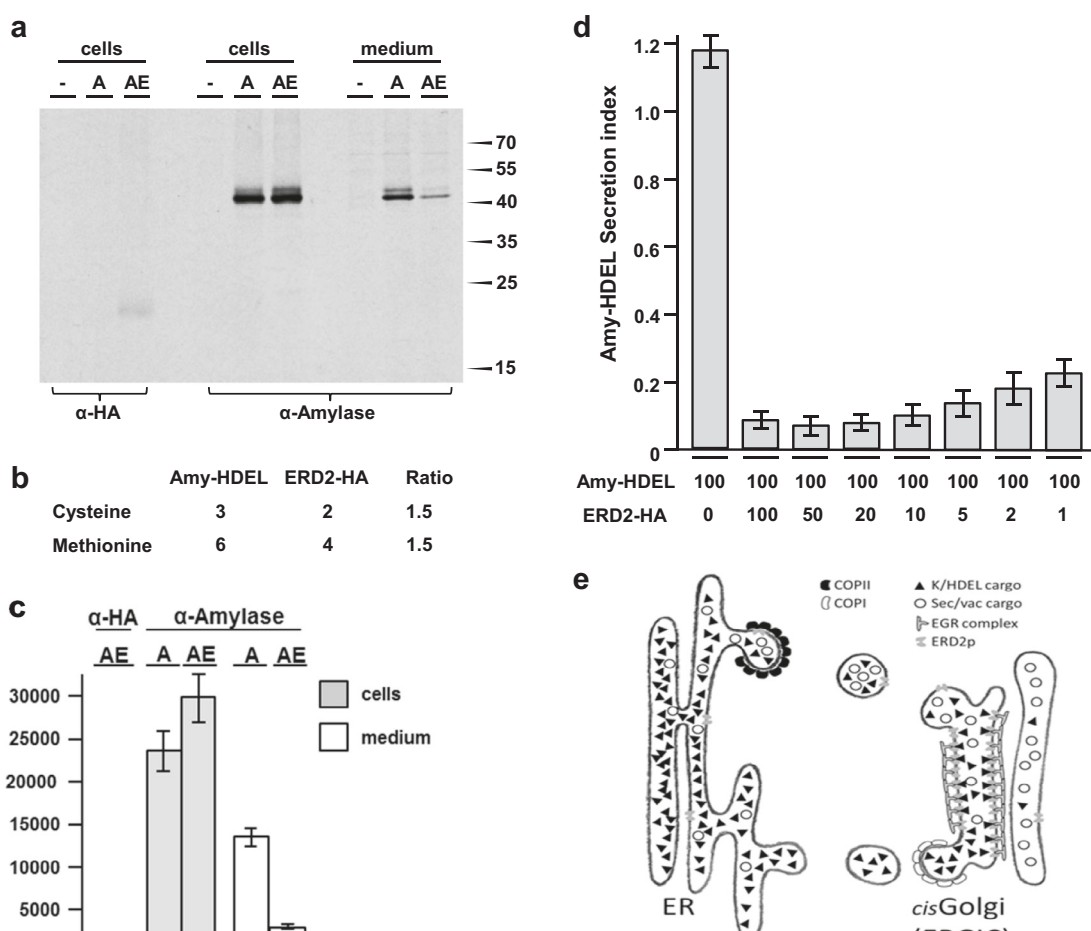

**Fig. 7 | Receptor-recycling cannot explain the observed receptor-ligand stoichiometry.** Transient expression in which both receptor and ligand were expressed from dual expression vectors harbouring GUS as reference marker for transfection efficiency and expressed at identical GUS units. **a** Immunoprecipitated ERD2-HA (α-HA) and co-expressed Amy-HDEL (α-Amylase) from mock-transfected cells (-), cells transfected with Amy-HDEL alone (A) and cells with Amy-HDEL and ERD2-HA (AE). Immunoprecipitated proteins were separated by SDS-page, followed by blotting on nitrocellulose and visualisation via phosphorimaging. Molecular weight markers are given on the right in kilo-daltons. Source data are provided as a Source data file. **b** Table showing the total number of cysteine and methionine residues in cargo and receptor. Relative radioactivity units measured for ERD2-HA by phosphorimaging must be multiplied by 1.5 to permit comparison with units from Amy-HDEL to permit calculation of relative number of molecules. Source data are provided as a Source data file. **c** Phosphorimaging quantification (arbitrary relative units) of signals from 3 different nitrocellulose blots as in (**a**) showing the radioactivity from transiently expressed ERD2-HA, Amy-HDEL (A) alone in cells and medium and Amy-

HDEL co-expressed with ERD2-HA (AE) in cells and medium. Error bars are standard errors from 3 biological replicas. **d** Retention of Amy-HDEL where the maximum receptor levels from panel **a** are co-transfected (second lane), followed by consecutive dilution of the receptor plasmid up to 100-fold (last lane). Notice that a strong reduction in Amy-HDEL secretion compared to the control (first lane) is still observed even after 100-fold dilution of the receptor plasmid (last lane). Error bars are standard deviations from at least 6 biological replicas. Source data are provided as a Source data file. **e** Schematic of the ER-Golgi interface, summarizing the current findings: K/HDEL cargo is several orders of magnitude more abundant than ERD2 and reaches the Golgi together with other cargo. A concentrated array of ERD2 molecules recognise K/HDEL cargo in the cis-Golgi/ERGIC lumen and releases them close to the COPI vesicle budding site. ERD2 itself is retained by an unknown ERD2-Golgi-retention (EGR) complex that interacts with the cytosolic di-leucine motif. The findings support a specific identity for the cis-Golgi/ERGIC and are difficult to reconcile with the cisternal progression model for intra-Golgi anterograde transport.

Since these findings question the classical receptor recycling model proposed three decades earlier and still considered to date[22], we have critically explored ERD2 properties to study its transport, the influence of ligands and sorting signals, and its ligand-transport capabilities. The results systematically point at a different mode of action, and raise further questions regarding the sub-compartmentalisation of Golgi cisternae and its cargo-sorting activity.

The recycling principle for protein sorting receptors has been popular in the field because it is thought to explain how few receptors can transport multiple ligands[7,9,31]. To explain the retrieval of soluble ER residents from the Golgi apparatus, a similar receptor recycling model was proposed[12], but this overlooks a fundamental difference: While vacuolar/lysosomal proteins are sorted only once from secreted proteins at the Golgi, ER residents are recycled back to where they

were originally synthesized. the ER[12]. Their perpetual escape to the Golgi would cause an endless "Sisyphean" task for ERD2. In addition, newly synthesized ER residents would add to the burden at each cycle. Given the high abundance of soluble ER residents[32], such odds would defeat even the most efficient receptor, and this conceptual problem has not been widely considered to date. If ERD2 accompanies its ligands to the ER in COPI vesicles, it would require "frequent-flyer" ER-to-Golgi export to outcompete the rate of ligand bulk flow. Here we have presented a series of results that strongly argue against the frequent flyer recycling principle.

Firstly, the lysine residues proposed to form an inducible KKXX-like signal for retrograde COPI transport[22] were completely irrelevant to human ERD2 activity when tested in our gain-of-function assay (Fig. 3b). This is in agreement with the fact that they deviate from the

established KKXX motif shown to be strictly C-terminal[33]. Secondly, equipping the ERD2 C-terminus with well-established COPII and COPI transport signals[28,29] causes ER localisation and inactivity, while mutating the canonical KKXX motif restores Golgi-localisation and activity (Fig. 5).

Two further experiments provide strong arguments for Golgi-retention of ERD2. FRAP analysis shows that the di-leucine motif is not required for ER export of the receptor but instead prevents its retrograde transport from the Golgi (Fig. 4). The signal is conserved in human ERD2 and promotes both activity (Fig. 3b) and Golgi-residency (Fig. 3d). Crucially, ligand-induced redistribution of ERD2 fusions to the ER only occurs when the di-leucine motif is deleted or masked (Fig. 4d, e). This explains previously published data[16,20,22,25] and shows that the discrepancy does not lie with what is observed, but with what is deemed biologically relevant. We conclude that ERD2-recycling is an artefact displayed by an inactive receptor fusion.

Perhaps the strongest argument against the recycling model came from the reciprocal approach, where inactive ERD2-YFP was supplemented with an alternative cis-Golgi retention signal[30]. This restored Golgi-residency as well as ERD2 activity in ligand-sorting despite the masking of its endogenous di-leucine motif (Fig. 6). Therefore, ERD2 has to reside at the Golgi in order to carry out its function. The fact that ERD2 expression under control of its own promoter in *P. patens* is extremely low (Fig. 1e) also correlates with a retention mechanism which establishes high steady-state levels solely at the cis-Golgi. FRAP analysis revealed no evidence for the presence of an active ER export signal and wild-type ERD2 ER-to-Golgi transport was not faster than that of a typical Golgi marker (Fig. 4a).

It is interesting that masking, mutating or even deleting the di-leucine motif did not lead to ERD2-leakage to post-Golgi compartments. ERD2 may thus contain 2 separate Golgi retention signals, the di-leucine motif to prevent ERD2 retrograde transport and a second signal to prevent anterograde Golgi-export. The latter remains a mystery[34], as does a full understanding of the mechanisms that maintain the spatial polar organisation of the Golgi stacks themselves[3].

Our measurement of the in vivo ERD2-ligand stoichiometry suggests a ratio higher than 1:100 (Fig. 7c). How can ERD2 avoid the "Sisyphus" paradox? Colloquially, the classic "frequent flyer" model can be illustrated by a taxi-driver who can take only a limited number of passengers. The driver can return to the collection point for another shuttle service, but while in transit any additional passengers have to wait. Our results suggest that ERD2 acts more like a "gatekeeper" at the cis-Golgi, illustrated by a bouncer who denies entry (Fig. 7e). Unlike a taxi driver, the bouncer holds its position at the gate and is able to repel a far greater crowd of individuals, including those making repeated attempts. This explains why the Golgi retention motif is crucial for ERD2-function and conserved in eukaryotes (Fig. 3c).

How could ERD2 avoid joining its ligands and maintain its gatekeeper position? Crucially, if ERD2 cycles between a ligand-bound and ligand-free form, this has to occur in the cis-Golgi itself. We could show that the GTPase ARF1 is stripped from Golgi-membranes and redistributes to the cytosol when ERD2 is overexpressed (Fig. 5a). This is accompanied by inhibition of constitutive secretion (Fig. 5b) and this effect is dependent on the presence of the di-leucine motif for ERD2 Golgi-retention (Fig. 5b). HDEL cargo can be detected in the cis-Golgi[35] and may represent cargo transiently associated with ERD2. ARF1-GAP recruitment[27] may help to retain ERD2 in a COPI-free zone and stimulate/drive ligand-release into an adjacent COPI-coated subdomain (Fig. 7e). It should be stressed that endogenous ERD2 levels are notoriously low and would only permit a small subdomain of the Golgi membrane to act as COPI-free zone, while recycling of K/HDEL proteins as well as KKXX proteins occurs via COPI carriers. For this reason, biochemical analysis of endogenous protein-protein interactions may prove challenging, and expressing higher levels of ERD2 may exacerbate interactions and cause artefacts.

The gatekeeper model for ERD2 stands apart from the recycling receptors controlling post-Golgi trafficking routes[4–6,11] and inspires further thoughts on the unknown origins of the ER and Golgi. When COPI was first associated with retrograde rather than anterograde transport[36,37] the cisternal progression model for Golgi polarity gained momentum[38]. At first sight our results seem to support the idea that the plant cis-Golgi could be a more permanent core-structure[39] in a similar nature to the mammalian ERGIC[40]. The possibility of ERD2 residency in Golgi entry core compartments (GECCOs)[41] cannot be excluded, but their presence here is unconvincing. ERD2 appears to readily distribute to the ER upon treatment with the drug Brefeldin A (Fig. 5c), which suggests that the mechanism of ERD2 retention in the Golgi is as dynamic as the organelle itself.

A precedent for a sorting facilitator that does not enter the transport carrier itself is Tango1, promoting collagen loading for ER export while effectively remaining behind in the ER membrane[42–44]. Collagen is one of the most abundant secretory cargos in fibroblasts, and a "taxi-driver" mechanism could be easily overwhelmed. An example of a Golgi-resident sorting component is RER1[45], which also depends on its C-terminus to accumulate in the Golgi apparatus, but it controls accumulation of membrane proteins in the ER. The similarity of ERD2 with sweet transporters[22], which are known to have multiple conformational states[46–48], may provide new clues to identify ancient transport mechanisms that could have initiated the formation of an ER-Golgi interface.

Future work should be devoted to experiments exploring (1) how ERD2 maintains its position at the Golgi apparatus without leaking beyond, (2) how it loads K/HDEL cargo for retrograde transport without joining and (3) how anterograde transport of non-ER proteins is achieved while the cis-Golgi retains its identity. The low concentration of endogenous ERD2 and the transient nature of protein–protein interactions involved in its sorting mechanism are likely to present a formidable challenge for the future.

## Methods

### Recombinant DNA construction
All plasmid constructs were created via standard techniques including PCR amplification, overlap PCR, QuickChange PCR mutagenesis, gene synthesis, oligonucleotide annealing, restriction digests, gel purification and ligation and the *E. coli* strain for plasmid replication was strain MC1061[49]. Supplementary Table 1 lists all plasmids and constructs used from earlier work[23] and new derivatives described below.

### Double vectors for stable transgenic ERD2 anti-sense lines
For generation of transgenic *N. benthamiana* lines, a nbERD2AB-antisense fragment followed by 3'nos polyadenylation signal was extracted from pJCA60[23] as a NcoI-HindIII, sub-cloned together with a second fragment (BamHI-NcoI, harbouring a 3'ocs polyadenylation signal followed by the CaM35S promoter), into *Agrobacterium tumefaciens* plant expression vectors pTJA15, pTFLA32 and pTMY1 between BamHI-HindIII. NbERD2AB-antisense mRNA is then transcribed from the strong constitutive CaM35S promoter and the second cassette encodes either ST-YFP-HDEL (pTJCA85), YFP-TM-ERD2b (pTJCA86) or ERD2b-YFP (pTRB29) under the control of weaker TR2 promoter.

### Targeted mutagenesis in *P. patens*
Establishment of the YFP-TM-PpERD2B-1/ΔPpERD2B-2 line: For construction of pYFP-TM-ERD2B-1, a fragment containing the YFP-TM sequence from pFLA30[23] was inserted directly between the end of the ERD2B-1 5'-UTR and the start codon of the ERD2B polypeptide coding sequence by overlap PCR, using primers (p1 + p2 + p3 + p4, Supplementary Fig. S1; Supplementary Table 2). The product, containing 1120 bp upstream of the coding region and 803 bp genomic DNA commencing from the start codon was cloned into an EcoRV site of pBluescript II KS⁻. For marker-free knock-in transformation of *P.*

*patens*, 15 µg of PCR amplified fragment (primers p5 and p6, Fig. 1; Supplementary Fig. 1; Supplementary Table 2) of pYFP-TM-ERD2B-1 containing 1056 bp and 760 bp of 5′ and 3′ flanking fragments, respectively, was mixed with 1 µg of supercoiled pMBL5 (GenBank Accession No. DQ228130) and used to transform *P. patens* by protoplast-PEG transformation[50]. Primary transformants grown on G418 containing medium were transferred to non-selective medium and initially screened for correct 3′-end targeting by PCR. Loss of the circular selection plasmid was confirmed by sensitivity to G418. The selected transformants were further screened for 5′-end targeting and concatenation events by PCR followed by Southern hybridisation to establish a correctly targeted, single-copy plant. Genomic DNA (2.5 µg) was digested with HindIII for electrophoresis and blotting onto nylon membrane. The probe comprising the 3′-end of YFP and the entire 3′-targeting fragment within YFP-TM-PpERD2B-1 (Supplementary Fig. 1, shaded box, primers p4 and p8, Supplementary Table 2) was labelled by PCR using 30% dTTP substituted with DIG-dUTP. The hybridisation and DIG detection was carried out in accordance with manufacturer's instruction.

To create an ERD2B-2 knock-out plant (Fig. 1; Supplementary Fig. 2; Supplementary Table 2), 5′- (956 bp: primers p9 and p10) and 3′-(935 bp: primers p11 and p12) targeting fragments were amplified from the genomic DNA located outside the ERD2B-2 coding region. These fragments were cloned either side of the 35S promoter-driven *nptII* cassette in pMBL5DLΔS[51]. The transgene containing 817 bp of 5′ and 868 bp of 3′ targeting sequence were bulk-amplified by PCR (primers p13 and p14) and used to transform *P. patens*::*YFP-TM-ERD2B-1*. The stable transformants were identified by two rounds of G418 selection. The targeted single-copy knock-out plants were confirmed by PCR followed by Southern hybridisation using an *nptII*-specific probe (primers p20 and p21). For the removal of the selection cassette, protoplasts of the *YFP-TM-ERD2B-1/ERD2B-2KO* plant were transiently transformed with 10 µg of the supercoiled rice actin promoter-driven Cre recombinase plasmid and allowed to grow on protoplast regeneration medium without selection. After 2 weeks, individual regenerants were sub-cultured onto fresh standard medium with and without selection. The marker removal was confirmed by PCR testing antibiotic-sensitive colonies for the absence of the selection cassette.

## ERD2 plasmids from different eukaryotic organisms

Gene synthesis services by Eurofins Genomics were used to obtain desirable coding sequences (CDS) and designed to be delivered in pUC57 vectors with restriction sites flanking both ends (ClaI and BamHI). Specific ERD2 sequences are shown in Supplementary Table 3.

For the bioassay analysis all CDS mentioned above were subcloned into an existing pJA31 vector, a double expression vector with a GUS internal marker previously described[23,52], via classical cloning utilising ClaI and BamHI restriction sites, substituting ERD2b gene and finally yielding TR2:GUS-35s:oiERD2 (*O. lucimarinus*), TR2:GUS-35s:acERD2 (*A. castellanii*), TR2:GUS-35s:piERD2 (*P. infestans*), TR2:GUS-35s:ccERD2 (*C. crispus*), TR2:GUS-35s:gsERD2 (*G. sulphuraria*), TR2:GUS-35s:hsERD2 (*Homo sapiens*), TR2:GUS-35s:hdERD2 (*H. dujardini*), TR2:GUS-35s:tpERD2 (*T. pseudonana*), TR2:GUS-35s:pgERD2 (*P. graminis*), TR2:GUS-35s:klERD2 (*K. lactis*), TR2:GUS-35s:tbERD2 (*T. brucei*), TR2:GUS-35s:yERD2 (*S. cerevisiae*).

## Human ERD2 derivatives

The construction of N-terminally tagged (YFP-TM-HsERD2) and C-terminally tagged (HsERD2-YFP) was carried out exactly as described previously for plant[23]. Primer BglII-hERD2 was used to introduce a BglII site and a short linker (Ile-Ser) to the HsERD2 N-terminus. Primer HsERD2-NheI was used to introduce an NheI site and a short linker (Ala-Ser-Ala) to the HsERD2 C-terminus. Both constructs were built and inserted either into the double expression vector with a GUS internal marker (pRB17 and pRB19) or into a T-DNA vector for Agrobacterium-mediated plant cell transformation for expression under the transcriptional control of the TR2 promoter (pTRB21 and pTJCA107).

Mutagenesis of human ERD2 was carried out using standard primers for quick-change mutagenesis as described in supplementary table 2 and implemented on the untagged human ERD2 construct in the dual expression vector for quantitative Amy-HDEL cell retention assays, followed by subcloning into the T-DNA vector encoding YFP-TM-HsERD2 to study the mutants via CLSM analysis.

## Epitope tagged plant and human ERD2

All C-terminal tags were inserted by trailer PCR using long antisense primers (Supplementary Table 2) annealing either with plant or human ERD2, followed by the trailer harbouring the relevant epitope coding region, a stop codon and restriction site XbaI, for PCR amplification in conjunction with cool35S (5′-CACTATCCTTCGCAAGACC-3′) followed by ClaI-XbaI insertion into either the double expression vector with a GUS internal marker for cargo sorting assays (6 plasmids, see Supplementary Table 1) or the T-DNA vector for CLSM analysis of the equivalent YFP-TM-ERD2 derivatives (6 further plasmids, see Supplementary Table 1).

## Deletions, hybrids and further derivatives of plant ERD2

To modify *Arabidopsis thaliana* ERD2b by deletions, chimeric hybrids and modified derivatives, a range of oligonucleotides were used either for direct trailer PCR or insertion of annealed primer pairs with sticky ends.

To replace the last 9 amino acids of ERD2 by the 9 amino acids of p24, the antisense primer ERD2b::p24tail was used combined with cool35S, followed by insertion into either the double expression vector with a GUS internal marker or the T-DNA vector for CLSM analysis. Mutagenesis of the di-lysine motif (KKSS) was done via Quick-change (Supplementary Table 2), followed by the same subcloning reactions.

To test the influence of an alternative Golgi-retention signal (LPYS), the primers LPYSsense and LPYSanti were used to anneal a DNA fragment with NcoI and ClaI compatible sticky ends (encoding MSNS**LPYS**VKDVHYDNAKFRQR) to replace the YFP coding region in pFLA30[23] cut out by the same enzymes, resulting in the LPYS-TM-ERD2 coding region (pRB22). LPYS-TM-ERD2 and the control TM-ERD2 (pFLA33) were provided with a C-terminal YFP in the same way as described for ERD2 before[23], resulting in double expression vectors with a GUS internal marker plasmids pRB25 and pRB26, and the T-DNA vectors pTRB25 and pTRB26.

## ERD2 constructs for re-distribution assays

To remove the di-leucine motif, the 5 last codons of ERD2b were removed by PCR amplification with primer ERD2bΔC5 anti, resulting in a coding region devoid of codons specifying the amino acids LQLPA, resulting in pTJCA88. The 35S-promoter driven ERD2b-YFP construct (pTJA10[23]) was re-constructed by replacing the 35S promoter by the weaker TR2 promoter by EcoRI-ClaI substitution, yielding pTMY1. To test the influence of overexpressed Amy and Amy-HDEL on several fluorescent ERD2 derivatives, the corresponding genes were recovered by HindIII digestions from pAmy and pAmy-HDEL, respectively, followed by blunting with Klenow and further digestion with EcoRI. Plant expression vectors pTJCA88, pTFLA32 and pTMY1 were prepared by SnaBI-EcoRI digests followed by dephosphorylation, resulting in the dual expression plant vector plasmids pTRB6, pTRB7, pTRB8, pTRB9, pTMY3, pTMY4 (Supplementary Table 1).

## Further fluorescent markers

The Golgi marker ST-RFP under the transcriptional control of the weak TR2 promoter in plasmid pTJA37 was described previously[23]. To generate an alternative Golgi-marker to specifically highlight *cis*-cisternae[30], the cytosolic N-terminus, the TM domain and a portion of the lumenal domain of MNS3 was amplified with MNS3 ClaI and MNS3

SalI from *N. benthamiana* protoplast cDNA, to replace the corresponding domains of ST-RFP[23], resulting in MNS3-RFP (pRB23). Subcloning into the T-DNA vector for Agrobacterium-mediated plant cell transformation for expression under the transcriptional control of the TR2 promoter resulted in pTRB23. ARF1-RFP was created by amplification of the ARF1 coding region as a NcoI-NheI fragment using primers ARF1-NcoI and ARF1-NheI and fused to an NheI-BamHI RFP fragment from pAW7 described earlier[11], to be expressed with the TR2 promoter.

## Plant material and transient gene expression

Sterile grown *N. benthamiana*[53] plants, protoplast preparation, electroporation and subsequent incubation and harvesting were done as described previously[23]. The tobacco leaf infiltration procedure with soil grown plants was done as described too[54].

## Confocal laser scanning microscopy

Forty-eight hours after infiltration, slides with tobacco leaf squares were prepared with tap water and imaged using an upright Zeiss LSM 880 Laser Scanning Microscope (Zeiss) with a PMT detector and a Plan-Apochromat 40x/1.4 oil DIC M27 objective using settings as described[23].

## FRAP analysis

Samples to be used in fluorescence recovery after photobleaching (FRAP) studies were pre-treated 48 h after infiltration to promote actin depolymerization and to stop Golgi movement. Small sections (0.5 × 0.5 cm) of the infiltrated leaves were removed and kept in a 12 μM solution of latrunculin B[54] (Cayman Chemical Co.) in water for one hour. Samples were then analysed via confocal laser scanning microscopy (CSLM). Golgi bodies showing both RFP and YFP fluorescence were selected as region of interest (ROI) shown as circles in Supplementary Fig. 4 and either kept as a control (closed arrow-head) or bleached (open arrow-heads), followed by recording of time-series to view the recovery of fluorescence in the bleached areas. Bleached Golgi bodies moving out of focus were detected via the disappearance of the red fluorescence and could therefore be discarded. To help the reader appreciate the rate of recovery, images in Supplementary Fig. 4 only show green fluorescence.

Zen 2.3 black edition (Zeiss) software was used to record pre- and post-bleached signals and to modulate laser beam intensity. Signals were sampled before bleach treatment using standard confocal setting as described before. Bleaching was achieved by scanning with high-intensity illumination of selected regions of interest (ROI) and every 30 s after bleaching with low-intensity illumination following recommendation of previously published protocol[55].

## Correlation analysis

Post-acquisition image processing was performed with the Zen 2.3 lite blue edition (Zeiss) and ImageJ ((http://rsb.info.gov/ij/)). Image analysis was undertaken using the ImageJ analysis program and the PSC co-localization plug-in[56] to calculate co-localization and to produce scatter plots as described before[11].

## Enzyme assays

Measurement of α-amylase activity and calculation of the secretion index (ratio of extracellular to intracellular enzyme activities) were done as described previously[23,52,57]. For GUS-normalised effector dose–response assays, the GUS enzyme essay was used. To reach best transfection practice (BTP), new dual expression plasmid preparations were first subject to transfection quality control by measuring the GUS activity after standard electroporations as described earlier[23]. The point at which GUS activity starts to approach a plateau is highly variable and cannot be predicted from the DNA concentration. A relative activity of 30 units (given in ΔOD per mL protoplast suspension and per hour enzyme incubation) prior to plateau conditions was deemed acceptable for BTP, although plateau values of up to 200 units can be reached. Plasmids that reached the GUS activity plateau with lower than 30 units were deemed unsuitable and were discarded. For GUS-normalised comparisons of different effector plasmids and for dose–response assays, plasmid doses were strictly determined volumetrically relative to the default plasmid concentration resulting in 30 GUS units. Generally lower doses were used and indicated as GUS equivalents, unless higher doses were used for overexpression (Figs. 5, 7).

## Generation of transgenic plants by leaf-disk transformation

*N. benthamiana* plants were obtained via Agrobacterium infection of leaf disks[58]. Selection of transformants was accomplished in MS medium supplemented with 3% sucrose and containing 100 μg/mL kanamycin and 250 μg/mL cefotaxime. Regenerated plants were analysed and scored by CLSM.

## In vivo labelling and immuno-precipitation

In vivo labelling of *N. benthamiana* protoplast suspensions was essentially done as described previously[59] with the following modifications: 4 repeats of standard electroporations[23] yielding 2.5 ml protoplast suspensions each were pooled together for each sample (mock, Amy-HDEL and Amy-HDEL + ERD2-HA). After 1 h rest in a standard 9 cm Petri Dish, the 10 mL pools were centrifuged at 100 rpm in conical tubes, followed by the removal of the dead cell pellet and the majority of the medium underneath the floating band of live cells. Protoplasts were then resuspended in a final volume of 2 mL TEX buffer and supplemented with 0.5 mL TEX containing 500 mCi/mL Pro-mix (70% 35S-methionine and 30% 35S-cysteine, Amersham Life Science), followed by incubation for 8 h at room temperature in the dark and harvesting of washed cell pellets and clear culture medium using the standard procedure. Culture medium was kept on ice for further work. Lysis of washed cell pellets was carried out by resuspending the washed cell pellet with 950 microlitres of ice-cold homogenization buffer (200 mM Tris-Cl, pH 8.0, 300 mM NaCl, 1% Triton X-100, 1 mM EDTA, and 2 mM phenylmethylsulfonyl fluoride). The homogenate was then centrifuged for 5 min in a minicentrifuge, and the supernatant was kept on ice for further work.

Immunoprecipitations were either carried out with 500 microlitres of culture medium or 200 microlitres of the cell lysis fraction, each representing 20% of the total labelled protoplast suspension, allowing direct calculation of secretion indices after quantification of signals from medium and cells. All manipulations were performed on ice or at 4 °C using ice-cold buffers. NET gel buffer (50 mM Tris-Cl, pH 7.5, 150 mM NaCl, 1 mM EDTA, 0.1% Nonidet P-40, and 0.02% NaN$_3$ and supplemented with 0.25% gelatin) was used to bring either the medium or the cell lysis fraction up to 1 mL. After centrifugation to remove any remaining debris, the supernatant was incubated on ice with an excess of either anti-HA antibody (1:1000 dilution, catalogue number GTX115044 from Genetex) or anti-amylase antibodies[35] (1:1000 dilution) for 1 h, allowing for the complete precipitation of all ERD2 or Amy-HDEL proteins after addition of protein A-sepharose and subsequent washing steps as described[59]. After the last wash, all liquid was removed from the washed protein A-sepharose pellets using a refined glass capillary. The pellets were then supplemented with thirty microlitres of SDS-PAGE loading buffer (200 mM Tris-Cl, pH 8.8, 5 mM EDTA, 1 M sucrose, 20 mM DTT, 2.5% SDS, 0.1% bromophenol blue) and the suspensions were incubated at 90 °C for 5 min. The sample was then centrifuged for 2 min in a minicentrifuge and 20 microlitres was separated on SDS-PAGE (10% strength), followed by electroblotting on nitrocellulose. Dried nitrocellulose sheets were analysed by phosphorimaging. Sample peak selection and detection were achieved by Aida version 4.14 and detector Fuji FLA-5000, respectively. Error bars are standard errors of three independent repeats. Arbitrary units of

pixel intensity are compared relative to other values within the same experiment, and corrected for the 2/3 ratio of amino acids methionine and cysteine in ERD2 relative to Amylase.

## Mammalian cell culture and transfections
The coding regions for the two fluorescent variants of plant and human ERD2 were inserted into the mammalian cell expression vector pcDNA3.1 under the transcriptional control of the CMV promoter, resulting in pJCA108 (YFP-TM-AtERD2), pJCA109 (AtERD2-YFP), pRB36 (YFP-TM-HsERD2), pRB52 (HsERD2-YFP).

HeLa CCL-2 cells were purchased from the American Type Culture Collection (Manassas, VA). These cells were cultured in Dulbecco's modified Eagle's medium (Thermo Fisher Scientific), supplemented with 100 units of penicillin/ml, 0.1 mg of streptomycin/ml, and 10% (v/v) fetal bovine serum (FBS). HeLa cells were transiently transfected with the plasmids indicated in the figure legends by using Lipofecta-mine 2000 reagent (Thermo Fisher Scientific) according to the manufacturer's instructions.

## Antibodies
For immunofluorescence assays, the following antibodies were used: the monoclonal mouse antibodies to GM130 (1:200 dilution; catalogue no.: 610822; clone 35/GM130; BD Biosciences), sheep polyclonal anti-TGN46 (1:400 dilution; catalogue no.: AHP500; Bio-Rad). Secondary antibodies conjugated to Alexa fluorophores were purchased from Thermo Fisher Scientific (donkey anti-mouse Alexa Fluor647 cat# A3157-1 and donkey anti-sheep Alexa Fluor594 cat# A11016, in both cases 1:1000 dilution).

## Immunofluorescence microscopy
Immunofluorescence was performed as previously described[60]. Briefly, cells were fixed for 15 min at RT with 4% (w/v) PFA in PBS. PFA-fixed cells were permeabilized with 0.01% (w/v) saponin in blocking solution (0.2% [w/v] pork skin gelatin in PBS) for 30 min at 37 °C, and double labelled with specific primary and secondary Alexa-conjugated antibodies. Cells were imaged on a Zeiss confocal laser-scanning microscope 780 (Zeiss). Post-acquisition image processing was performed with Fiji/ImageJ software (https://imagej.net/software/fiji/).

## Reporting summary
Further information on research design is available in the Nature Portfolio Reporting Summary linked to this article.

## Data availability
This project has generated a large number of raw data that require standard manipulations such as subtracting a blank OD, calculating enzyme activities, calculating ratios between medium and cell samples, calculating averages, standard deviations and standard errors, all of which can be made available by the Lead contact upon reasonable request. For Fig. 2a, the raw data for cells and medium, as well as the calculation of the secretion index can be found in the Source data. Individual averages from separate biological replicas for Figs. 3b, 3f, 5b, 5e, 6c, 7d and Supplementary Figures 3a, 3b, 3c, 6a, 6b, are included in Source data. Uncropped gel images, blots, autoradiographs and FRAP data (Figs. 1, 4a, 7; Supplementary Fig. 4a) can also be found in Source data. Source data are provided with this paper.

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

## Acknowledgements

The work in this article was in part supported by the European Union (projects HPRN-CT-2002-00262 and LSH-2002-1.2.5-2), the Biotechnology and Biological Sciences Research Council (BBSRC) project nr. BB/D016223/1, The Leverhulme Trust (F/10 105/E) and the São Paulo Research Foundation (FAPESP), Brazil (grant no.: 19/02418-9 and 19/26119-0). Jonas C. Alvim is grateful for a PhD scholarship awarded from the Conselho Nacional de Desenvolvimento Científico e Tecnológico – Brasil (CNPq 201192/2014-4). Robert M. Bolt is grateful for a Faculty of Biological Sciences PhD studentship funded by the University of Leeds. Juan O. Concha is grateful for a FAPESP Doctoral Scholarship (grant no.: 20/11900-6). Reese's Cups (Hershey Company, 1025 Reese Ave, Hershey, PA 17033) is thanked for providing food for thought. Jack Ranger and Nicoletta Bencka are thanked for contributing to the construction of recombinant plasmids (Supplementary Table 1). David Gershlick is thanked for scientific discussion and critically reading the manuscript.

## Author contributions

J.D. conceptualised the project with valuable input from J.C.A. and R.B. Experiments were performed and data was collected by J.C.A., R.B., J.A.,

Y.K., A.C., F.S.A., J.C., L.d.S., M.H., D.H. and J.D. Data analysis, data interpretation and figures were generated by J.C.A., R.B., J.A., A.C., L.d.S. and J.D. The manuscript was written by J.D. with contributions from J.C.A. and R.B.

## Competing interests

The authors declare no competing interests.
