## [Peer Review File · Nature Communications]

REVIEWER COMMENTS

Reviewer #1 (Remarks to the Author):

This paper is a follow-up of the Plant Cell paper published in 2018, which for the first time showed a novel di-leucine motif that is necessary to retain the receptor ERD2 in the plant Golgi. Here, the authors further show that this motif retains ERD2 in the Golgi by preventing its retrograde transport from the Golgi to the ER, which renders ERD2 inactive.

I find this work very appealing and elegant and it is obvious that the authors have made a great effort to prove the importance of the di-leucine motif in maintaining ERD2's Golgi residency and biological function. I recommend accepting the manuscript with minor revision.

General remarks:

I am not familiar with the journal's policy regarding microscopy images, but generally, red images should be avoided and colored in magenta since 1 in 4 males suffers from red-green color blindness.

All microscopy images should be provided with scale bars.

Main text:

line 30: The authors may consider using the more general term "transport carriers" instead of "transport vesicles".

line 84: I think that the sentence "Figure 1E, panel 2, white arrow heads" in brackets should be: "Figure 1E, panel 2, white arrowheads pointing at Golgi stacks".

line 90: briefly explain the assay. Not everyone (especially non-plant cell biologists) might be familiar with the amy secretion assay.

line 122: Is the "minor effect" statistically significant?

line 123: The authors mention "predominant Golgi localization. What else was observed? ER?"

line 135: Did the authors do a BFA treatment to test whether ERD2 is retained on Golgi membrane remnant structures such as AtCASP, AtSYP31 or AtMNS3? Might be another confirmation that the di-leucine motif acts as a "real" Golgi retention signal and provides ERD2 with the task of an essential gate-keeper at the entry point to the Golgi.

line 161 and 162: ST-YFP instead of ST-RFP

line 187: The authors conclude that the ER export signal in the p24 C-terminus compensates for the lack of the di-leucine Golgi retention signal. I think that this is not the case; the two signals have

completely different functions. The p24 signal simply transports ERD2 from the ER to the Golgi, which is then retained there by another mechanism in the absence of the di-leucine signal. In fact, there must be another retention mechanism in place since the LLGG mutant of ERD2 is still present in the Golgi (and only partially in the ER). The many transmembrane domains of ERD2 might play a role in Golgi localization as well.

line 149: There are no panels b3, b4 in Suppl. Fig. 5B.

Figures:

Figure 2H: In the confocal image with human ERD2 HA, some Golgi stacks look a little bit smeared. Did those Golgi bodies move during image acquisition?

Figure 3A: ERD2 LLGG fluorescence recovers faster and to a larger extent than the wildtype due to its partial ER localisation. In order to find out whether the mutant recovers faster due to de novo synthesized protein in the ER or whether ERD2 LLGG cycles from the Golgi to the ER and back to the Golgi, it would be worth repeating this FRAP experiment in the presence of cycloheximide. CHX aborts protein synthesis and the authors could confirm that fluorescence is replenished by de novo synthesized protein from the ER.

Figure 4A: What happens when TR2:ST-YFP + 35S:deltaC5 are co-expressed with ARF1-RFP? Does ARF1 remain Golgi-localised? Also, a co-IP of ERD2 and its mutant version, respectively, with ARF1 (via the fluorescent tag or HA) would be desirable in order to confirm an interaction (or not) with ARF1. Also, I would like to see what happens to the localization of ERD2 in the experiment, where TR2:ST-YFP + 35S:ERD2 are co-expressed with ARF1-RFP. Does ERD2 remain on Golgi bodies?

Figure 4C to E: Please use the same names in all three figures for clarity.

Figure 4E: Regarding the images showing co-localization of KKSS with ST-YFP: the color on colocalizing Golgi bodies only shifts in the same direction. Is this chromatic aberration? Maybe the authors can find different images.

Figure 6A and C: I am missing a proper legend. Is "m" the same as "E"? I presume that "m" stands for mock and is the same as "E" which is ERD2 expression without Amy-HDEL. Please provide a clear legend that is self-explanatory.

Suppl. Fig.4A: In Brandizzi et al. Plant Cell 2002, ST-GFP recovered to 80-90% of pre-bleach within 5 min. In Schoberer et al. Traffic 2009, the two Golgi-localised glycosylation enzymes GnTI and GalT also recovered to 80-100% within 5 min. Here, ST-YFP recovered to 60% within 10 min, which considerably differs from previously published data. Any comment on the reason for this difference?

Discussion:

The authors do not speculate on why the mutation places ERD2 partially in the ER. Does the mutated ERD2 protein pile up in the ER due to inefficient ER export?

How does ERD2 get to the Golgi before it is retained there? Did the authors look for an ER export or Golgi targeting signal within the ERD2 sequence?

Where could ERD2 be turned over? Have the authors ever silenced any of the COPI subunits to see whether ERD2 lights up in a post-Golgi compartment such as the vacuole?

Also, I would like the authors to speculate on the biological nature of ERD2 retention in the Golgi with regard to the di-leucine motif. Do the authors think that the motif potentially interacts with another protein/complex or do they prefer the idea of a protein-lipid interaction? Since this paper is a lot about this motif, some speculation would be desirable.

Reviewer #2 (Remarks to the Author):

This is a sensational paper. It is a follow-up of the authors' previous work in 2018, which described that the K/HDEL receptor ERD2 lost its function when either its N- or C-terminal was masked by XFP fusion and, in its native form, it predominantly localized to the Golgi. In the present paper, the authors have extended their study along this line and concluded that ERD2 does not recycle between the Golgi and the ER. Carefully engineered active ERD2 derivatives reside in the Golgi almost exclusively. The authors have shown that the C-terminal di-leucine signal is critical for the function and its mutations abolish the Golgi residency and the ability to retain amylase-KDEL in the ER. Importantly, the turnover of the receptor as examined by FRAP is not inhibited by the mutation but is rather accelerated, suggesting that the rapid recycling is not the mechanism of the ERD2 function. Furthermore, the ligand-induced relocation of the receptor to the ER is only seen for function-less ERD2 mutants. The addition of an ER export signal or a Golgi-retention signal to mutant KDEL partially restores its function.

It is an amazing and provocative story, because the rapid cycling model of the retrieval receptor has been favored for a long time in the field. The authors biochemically estimated the molecular ratio of ligand vs receptor as greater than 100 to 1, which would impose the endless "Sisyphus" task to the receptor if it needs to act as a frequent flyer. Instead, the authors propose that the receptor plays a bouncer or gatekeeper role to block entry of the ligands into the cis-Golgi. From the finding that the overexpression of ERD2 causes release of Arf1 GTPase into cytosol, they suggest a model that ERD2 is in a COPI-free zone so that it avoids joining the Golgi-to ER retrieval.

Experiments are well designed and the data obtained are mostly clear and convincing. As the findings of this work are very impactful to the field, I would support its publication. However, there are several points that need to be clarified before acceptance as listed below. I have a few

suggestions for the authors to strengthen their claims. Furthermore, I have to say that the manuscript should be carefully revised to correct errors and shortage of appropriate explanations. I had a hard time to understand what the authors mean at several places.

1. The FRAP experiment and its results are very important in this paper. However, the interpretation of the data needs some caution. The recovery of fluorescence signals could be explained by various ways. ER-Golgi cycling is one of the possible causes, but in addition to that, several other sources could contribute to the recovery, for example, interexchange between Golgi stacks, post-Golgi compartments, or even small vesicular pools which are not detectable under conventional microscopes. The exchange between Golgi stacks may occur via such vesicular communications. To argue that the fluorescence recovery is indeed from the ER, the authors may want to look at the effect of inhibition of ER-Golgi anterograde traffic by, for example, the dominant negative mutant of Sar1.

2. Related to the above point, the authors described in the paper that ERD2 leakage to post-Golgi compartments was undetectable, but I wonder how carefully the authors examined possible relocation of the receptor to post-Golgi compartments, especially when mutant versions were expressed.

3. What recognizes the di-leucine motif is also an important issue. Apparently, it does not function like the conventional Golgi-to-ER retrieval signal. Then the obvious question is what binds it. With COPI as a negative control, can the authors look at its biochemical property whether it binds COPII, Vps74/GOLPH3 (if its homolog exists in plants), or any other coat and adaptor proteins?

Others are relatively minor points but should be carefully checked and corrected.

4. L67-70. This explanation is awful. I couldn't understand the meaning until I went back to the previous 2018 paper.

5. L120-121. Supplemental figure 3C. Where is it?

6. L195-202. The figure referred to here should be Figure 5, not 6.

7. L475-487. This part is awkward. It looks like a mixed-up version of copy and paste. In L487, what do black arrow-head and white arrow-head mean?

8. L768-770. This is awful again. From this explanation, nobody can understand the difference between B1, B2 and B3. I was totally at a loss at the first glance.

9. L813. Supp Figure 3. Where is it?

10. L835-838. The explanation about FRAP is too poor in the legend. How did the authors choose ROIs? How long time were they bleached for? It is not good to force readers to look into Materials and Methods to get essential information.

11. L895-914. Again, this is a terribly unkind figure legend. What are m, A and AE!? I went back to the main text and Material and Methods section repeatedly but couldn't find the answer and gave up.

12. L982-994. I am not sure Supplementary Figure 3 is relevant in this paper.

Reviewer #3 (Remarks to the Author):

Denecke and co-workers studied the dynamics of the K/HDEL receptor in plant cells.

They first showed that ERD2 fused with an additional N-terminal transmembrane domain and a YFP is able to complement the loss of endogenous ERD2 in both *Nicotiana benthamiana* and *P. patens*. Overexpression of 9 out of 12 different ERD2 from different sources reduced the secretion of amy fused with HDEL or KDEL. YFP-TM-HsERD2 localised exclusively in the Golgi complex, whereas addition of YFP to the C-terminal showed dual ER and Golgi localisation and was unable to retain amy-HDEL intracellularly.

Analysis of several C-terminal residues of ERD2 showed that K206 and K207 are dispensable for amy-HDEL retention, while leucines at positions -2 and -3 are required. The addition of small tags to the C-terminus of ERD2 also impaired the receptor's ability to retain amy-HDEL, except for HA tag. Note that retention of amy-HDEL correlates with the exclusive presence of ERD2 in the Golgi. The deletion

of the two leucines did not affect ER exit, but most likely the retention of ERD2 in the Golgi. Furthermore the deletion of the last 5 amino acid impaired the receptor ability to retain amy-HDEL.

To examine the role of COPI/COPII on ERD2 localisation, they generated a chimera in which the C-terminus of ERD2 was replaced with that of p24. This chimeric receptor was retained in the ER and was unable to prevent amy-HDEL secretion unless the two terminal lysines were mutated. In addition, introduction of a Golgi retention signal into a non-functional TM-ERD2-YFP restored Golgi localization and partial retention activity of amy-HDEL.

Despite similar plasmid transfection levels, the expression of ERD2-HA was much lower than that of amy-HDEL, but still sufficient to reduce amy-HDEL secretion. Quantitative analysis indicated that 4.5 amy-HDEL molecules are retained in the cells per ERD2. Although, an accurate dose-response curve suggested that 2 molecules of ERD2 can retain up to 100 molecules of amy-HDEL.

In summary, this study using a number of artificial ERD2 constructs showed that perturbation of the K/HDEL receptor alters its intracellular distribution and ability to retain an overexpressed ligand (amy-HDEL). It also showed an association between the localisation of the receptor in the Golgi and its ability to retain amy-HDEL. While it is unclear what is happening when ERD2 has a mixed ER-Golgi distribution: in this case, although amy-HDEL promotes the redistribution of the receptor to the ER (Fig. 3D), it is not retained within the cells (Fig. 4C).

Experiments supporting a role of C-terminal leucines in interfering with COPI-dependent retrograde transport of the receptor are insufficient and rather indirect. For example, Co-IP experiments should be performed to understand how the ERD2 interactome changes by removing these leucines and thus possibly clarify the relationship with COPI. We believe that to date these experiments only suggest a role for leucines in slowing down retrograde receptor transport. The fact that introducing the C-terminal region of p24 to ERD2 alters the distribution and function of the chimeric proteins may not be surprising. We cannot exclude an important effect on the three-dimensional organisation of the receptor.

In our opinion, the main problem is that the experiments do not support the message the manuscript wants to convey. The data presented do not demonstrate that endogenous ERD2 is involved in the recovery of endogenous K/HDEL proteins while remaining in the Golgi complex, and over 20 years of evidence suggests otherwise. We do not know which model is correct, but the evidence provided, based only on the overexpression of modified proteins, is not sufficient to change the common paradigm.

Minor points

The manuscript is rather difficult to follow, most of the acronyms are undefined and the experiments are inadequately described, assuming that the reader is perfectly familiar with the previous manuscript by the same author (Fernanda et al *The Plant Cell* 2018), but this may not be the case for the wide readership of nature communication.

REVIEWER COMMENTS

Reviewer #1 (Remarks to the Author):

This paper is a follow-up of the Plant Cell paper published in 2018, which for the first time showed a novel di-leucine motif that is necessary to retain the receptor ERD2 in the plant Golgi. Here, the authors further show that this motif retains ERD2 in the Golgi by preventing its retrograde transport from the Golgi to the ER, which renders ERD2 inactive.

I find this work very appealing and elegant and it is obvious that the authors have made a great effort to prove the importance of the di-leucine motif in maintaining ERD2's Golgi residency and biological function. I recommend accepting the manuscript with minor revision.

General remarks:

I am not familiar with the journal's policy regarding microscopy images, but generally, red images should be avoided and colored in magenta since 1 in 4 males suffers from red-green color blindness.

The corresponding author is colour-blind and believes that for the relatively straightforward data the standard red-green is the best compromise to cater for a majority of readers. Blue-green is the preferred choice for colour-blind viewers but the blue signal is harder to see against a black background for the majority of viewers. Magenta does not help much for colour-blind viewers because it causes problems to distinguish the merge because magenta is already rather bright.

All microscopy images should be provided with scale bars.

Agreed these have now been included

Main text:

line 30: The authors may consider using the more general term "transport carriers" instead of "transport vesicles".

This has been considered and changed when appropriate.

line 84: I think that the sentence "Figure 1E, panel 2, white arrow heads" in brackets should be: "Figure 1E, panel 2, white arrowheads pointing at Golgi stacks".

The figure has been modified and the description has improved

line 90: briefly explain the assay. Not everyone (especially non-plant cell biologists) might be familiar with the amy secretion assay.

We agree, this was written in a far too condensed form. We have expanded this description to make it more accessible for a wide readership.

line 122: Is the "minor effect" statistically significant?

Yes the HA-tagged ERD2 is clearly less active compared to untagged ERD2, but the effect is very small compared to the strong reduction in activity when other tags were used (c-myc, FLAG). The

dose response in supplementary data shows that HA tagging is not neutral, but the effect is really small.

line 123: The authors mention “predominant Golgi localization. What else was observed? ER?”

We have added “with almost none detected in the ER, unless very high detector gain settings are used”. The choice of the term “predominant” was inspired by careful thinking, as we do not wish to categorically exclude localisation elsewhere (perhaps in transit) but we simply cannot detect it in post Golgi locations.

line 135: Did the authors do a BFA treatment to test whether ERD2 is retained on Golgi membrane remnant structures such as AtCASP, AtSYP31 or AtMNS3? Might be another confirmation that the di-leucine motif acts as a “real” Golgi retention signal and provides ERD2 with the task of an essential gate-keeper at the entry point to the Golgi.

This was an excellent suggestion and we have addressed this experimentally. We were unable to see good evidence for remnant structures labelled with fluorescent ERD2, suggesting that ERD2 cannot escape the progressive mixing of Golgi membranes with ER membranes. As this is interesting as part of the characterisation, we decided to include the data.

line 161 and 162: ST-YFP instead of ST-RFP

done

line 187: The authors conclude that the ER export signal in the p24 C-terminus compensates for the lack of the di-leucine Golgi retention signal. I think that this is not the case; the two signals have completely different functions. The p24 signal simply transports ERD2 from the ER to the Golgi, which is then retained there by another mechanism in the absence of the di-leucine signal. In fact, there must be another retention mechanism in place since the LLGG mutant of ERD2 is still present in the Golgi (and only partially in the ER). The many transmembrane domains of ERD2 might play a role in Golgi localization as well.

This point is important and has stimulated us to add further data. We wholeheartedly agree that the p24 signals are completely distinct from the ERD2 di-leucine signal. We also agree that the LLGG mutant, the deltaC5 lacking the di-leucine signal and the ERD2-YFP with a masked di-leucine motif are all still detectable at the Golgi. Export from the Golgi to distal locations such as the plasma membrane, the endosomes or even the vacuole membrane must be prevented by another mechanism, may be restricted by the length of the transmembrane domains. However, the signals at the Golgi are deceptive. The ER network comprises a much larger membrane surface, therefore the weaker fluorescence at the ER is simply caused by the protein diffuse in a much larger structure compared to the small Golgi stacks. In addition, we have made it more clear that low expression appears to favour ER localisation of a non-functional ERD2 derivative such as ERD2-YFP. This is illustrated with ERD2-YFP in figure 1 and supplementary figure 4? We would argue that the microscopy underestimates how much of ERD2-YFP is really in the ER. Since we have now added experimental results showing YFP-TM-ERD2 and ERD2-YFP in HeLa cells (Figure 2B) showing a nicer overview of entire cells, it is clear that masking the di-leucine motif leads to the majority of the ERD2 derivative to be erroneously localised in the ER.

The ERD2 fusion with the wild type p24 C-terminus is fully ER localised which demonstrates the dominance of the canonical KKXX motif and also that this is incompatible with ERD2 function. When we mutated the KKXX motif (ERD2-P24KKSS), the fusion is remarkably active in sorting HDEL-cargo and is also really quite restricted to the Golgi apparatus. To test if this is due to Golgi-retention, we carried out the ligand-redistribution assay and saw that it was redistributed to the ER. The term "compensation" was chosen because it is not mediating Golgi-retention, but that faster ER export via a COPII signal compensates for the lack of the di-leucine motif (or suppresses the lack of the dileucine motif)

line 149: There are no panels b3, b4 in Suppl. Fig. 5B.

This error has been corrected

Figures:

Figure 2H: In the confocal image with human ERD2 HA, some Golgi stacks look a little bit smeared. Did those Golgi bodies move during image acquisition?

Yes, plant Golgi bodies are highly mobile and it is hard to avoid during high resolution imagery. The main purpose of the figure was to show that the HA-fusion is not detected in the ER, unlike the fusions with FLAG and c-myc.

Figure 3A: ERD2 LLGG fluorescence recovers faster and to a larger extent than the wildtype due to its partial ER localisation. In order to find out whether the mutant recovers faster due to de novo synthesized protein in the ER or whether ERD2 LLGG cycles from the Golgi to the ER and back to the Golgi, it would be worth repeating this FRAP experiment in the presence of cycloheximide. CHX aborts protein synthesis and the authors could confirm that fluorescence is replenished by de novo synthesized protein from the ER.

The main purpose of the FRAP experiment was to test if the di-leucine motif is a fast ER export signal. The answer is most definitely no, because if it were the case the mutant would have shown slower FRAP. The fact that the mutant actually recovers slightly faster is a small detail that we are happy to speculate on. The ER membrane surface is huge in comparison to the small Golgi bodies, so the pool of YFP-TM-ERD2LLGG is very high at time 0 of our FRAP. The presence of cycloheximide will only have a minor effect and from the Moss experiments we know that ERD2 promoters are very weak. We did not wish to spend more time on the characterisation of non-functional ERD2-fusions.

Figure 4A: What happens when TR2:ST-YFP + 35S:deltaC5 are co-expressed with ARF1-RFP? Does ARF1 remain Golgi-localised?

This is an excellent point and we have included this control. Essentially, 35S:ERD2deltaC5 has no effect on ARF1-RFP.

Also, a co-IP of ERD2 and its mutant version, respectively, with ARF1 (via the fluorescent tag or HA) would be desirable in order to confirm an interaction (or not) with ARF1. Also, I would like to see what happens to the localization of ERD2 in the experiment, where TR2:ST-YFP + 35S:ERD2 are co-expressed with ARF1-RFP. Does ERD2 remain on Golgi bodies?

We totally agree that protein-protein interactions between wild type and mutant receptors (i.e. delta C5 or LLGG mutants) will pave the way towards identifying the mechanism of how ERD2 avoids entering COPI carriers but this is an ongoing project and outside the scope of this manuscript.

The BFA experiment with ERD2 shows it does not persist at Golgi stacks, so is unlikely to behave differently when it causes the phenotype itself. YFP-TM-ERD2 alone can also cause the beginning of a BFA effect when overexpressed, which is first manifested by Golgi stack clustering, followed by appearance in the ER and nuclear envelope. As endogenous ERD2 is expressed at very low levels, this effect would not be very prominent in nature, but it sheds light on the first place to look if we want to understand how ERD2 avoids the round trip to the ER. The mechanism will be explored in a follow-up paper.

Figure 4C to E: Please use the same names in all three figures for clarity.
This suggestion has been taken on board.

Figure 4E: Regarding the images showing co-localization of KKSS with ST-YFP: the color on colocalizing Golgi bodies only shifts in the same direction. Is this chromatic aberration? Maybe the authors can find different images.

We agree that the microscopy was compromised by faulty equipment and have done the microscopy again. We have obtained better quality images and replaced them.

Figure 6A and C: I am missing a proper legend. Is “m” the same as “E”? I presume that “m” stands for mock and is the same as “E” which is ERD2 expression without Amy-HDEL. Please provide a clear legend that is self-explanatory.

Agreed, the legend and figure has been improved accordingly.

Suppl. Fig.4A: In Brandizzi et al. Plant Cell 2002, ST-GFP recovered to 80-90% of pre-bleach within 5 min. In Schoberer et al. Traffic 2009, the two Golgi-localised glycosylation enzymes GnT1 and GalT also recovered to 80-100% within 5 min. Here, ST-YFP recovered to 60% within 10 min, which considerably differs from previously published data. Any comment on the reason for this difference?

These differences do not surprise us because cell dynamics is greatly affected by the external environment, such as temperature and also the physiology of the leaves. Individual research groups use different microscopes and settings, other plants and growth conditions and even the temperature in the microscope room can vary enormously from laboratory to laboratory. The recovery time can only be interpreted relative to the control within the same experiment. Most importantly, Brandizzi and Schoberer used constructs expressed from the strong 35S promoter, whilst we used the weaker TR2 promoter to avoid overexpression artefacts. Invariantly this means that during the fluorescence recovery, the bleaching effect of continuous scanning will be greater in our case, leading to a slower recovery.

Discussion:

The authors do not speculate on why the mutation places ERD2 partially in the ER. Does the mutated ERD2 protein pile up in the ER due to inefficient ER export?

How does ERD2 get to the Golgi before it is retained there? Did the authors look for an ER export or Golgi targeting signal within the ERD2 sequence?

Both points were explained but we admit that the first submission was a much condensed version which we were only too happy to expand during the review process. We clearly conclude that the mutant appears in the ER because a Golgi-retention motif is missing, and that ERD2 does not have a frequent flyer ER export signal but simply reaches the Golgi by bulk flow.

Where could ERD2 be turned over? Have the authors ever silenced any of the COPI subunits to see whether ERD2 lights up in a post-Golgi compartment such as the vacuole?

ERD2 is normally synthesised at very low levels and would therefore require overexpression to permit analysis of disposal routes. However we know overexpression causes a BFA-like effect at the ER-Golgi interface so this may not work either. We agree this is an interesting point to be addressed in the future.

Also, I would like the authors to speculate on the biological nature of ERD2 retention in the Golgi with regard to the di-leucine motif. Do the authors think that the motif potentially interacts with another protein/complex or do they prefer the idea of a protein-lipid interaction? Since this paper is a lot about this motif, some speculation would be desirable.

We can certainly speculate and suspect that it may interact with a prominent Golgi-matrix protein. But even then its behaviour at the Golgi is expected to be more dynamic because it needs to cycle between ligand-free and ligand-bound configurations within the same Golgi membrane and segregate between subdomains. Here we simply state that it appears to avoid COPI carriers and certainly does not contain a COPI signal itself, but identifying the various binding partners will have to be addressed in future experiments.

Reviewer #2 (Remarks to the Author):

This is a sensational paper. It is a follow-up of the authors' previous work in 2018, which described that the K/HDEL receptor ERD2 lost its function when either its N- or C-terminal was masked by XFP fusion and, in its native form, it predominantly localized to the Golgi. In the present paper, the authors have extended their study along this line and concluded that ERD2 does not recycle between the Golgi and the ER. Carefully engineered active ERD2 derivatives reside in the Golgi almost exclusively. The authors have shown that the C-terminal di-leucine signal is critical for the function and its mutations abolish the Golgi residency and the ability to retain amylase-KDEL in the ER. Importantly, the turnover of the receptor as examined by FRAP is not inhibited by the mutation but is rather accelerated, suggesting that the rapid recycling is not the mechanism of the ERD2 function. Furthermore, the ligand-induced relocation of the receptor to the ER is only seen for function-less ERD2 mutants. The addition of an ER export signal or a Golgi-retention signal to mutant KDEL partially restores its function.

It is an amazing and provocative story, because the rapid cycling model of the retrieval receptor has been favored for a long time in the field. The authors biochemically estimated the molecular ratio of ligand vs receptor as greater than 100 to 1, which would impose the endless "Sisyphus" task to the receptor if it needs to act as a frequent flyer. Instead, the authors propose that the receptor plays a bouncer or gatekeeper role to block entry of the ligands into the cis-Golgi. From the finding that the overexpression of ERD2 causes release of Arf1 GTPase into cytosol, they suggest a model that ERD2 is in a COPI-free zone so that it avoids joining the Golgi-to ER retrieval.

Experiments are well designed and the data obtained are mostly clear and convincing. As the findings of this work are very impactful to the field, I would support its publication. However, there are several points that need to be clarified before acceptance as listed below. I have a few suggestions for the authors to strengthen their claims. Furthermore, I have to say that the manuscript should be carefully revised to correct errors and shortage of appropriate explanations. I had a hard time to understand what the authors mean at several places.

We thank the reviewer for the supportive comments and agree that the paper was written in a condensed format. We have taken the time to explore the more generous word limit of Nature Communications to improve clarity and force of message.

1. The FRAP experiment and its results are very important in this paper. However, the interpretation of the data needs some caution. The recovery of fluorescence signals could be explained by various ways. ER-Golgi cycling is one of the possible causes, but in addition to that, several other sources could contribute to the recovery, for example, interexchange between Golgi stacks, post-Golgi compartments, or even small vesicular pools which are not detectable under conventional microscopes. The exchange between Golgi stacks may occur via such vesicular communications. To argue that the fluorescence recovery is indeed from the ER, the authors may want to look at the effect of inhibition of ER-Golgi anterograde traffic by, for example, the dominant negative mutant of Sar1.

The purpose of the FRAP experiment was to test if the di-leucine signal is a fast ER export signal and required for receptor arrival at the Golgi. If this were the case, FRAP should have been much slower in the mutant. This was not the case and therefore we can rule this out.

That recovery of the mutant is actually a little faster was a detail, and since the mutant has lost biological activity any further analysis was not really a priority. We speculate that the most likely source of recovery is the increased pool of YFP-TM-ERD2LLGG in the ER because this is the main difference between the wild type and the mutant. We would not want to go further than that and we fully agree that it is speculative.

2. Related to the above point, the authors described in the paper that ERD2 leakage to post-Golgi compartments was undetectable, but I wonder how carefully the authors examined possible relocation of the receptor to post-Golgi compartments, especially when mutant versions were expressed.

ERD2 fusions have either been detected in the Golgi or in both the Golgi and the ER. No punctate structures of any kind other than Golgi bodies were seen, and there was no labelling of the plasma membrane or the tonoplast either. We speculate that the ERD2 core structure employs another mechanism to prevent Golgi-export to distal compartments, but would not wish to be too prescriptive on that front. Also, the appearance of ERD2 in the ER when the di-leucine motif is mutated, deleted or masked is quite substantial, and this is particularly well illustrated when comparing YFP-TM-ERD2 and ERD2-YFP in HeLa cells which provide a nice overview of entire cells. This underlines that the primary purpose of the di-leucine motif is prevent Golgi-to-ER recycling of ERD2.

3. What recognizes the di-leucine motif is also an important issue. Apparently, it does not function like the conventional Golgi-to-ER retrieval signal. Then the obvious question is what binds it. With

COPI as a negative control, can the authors look at its biochemical property whether it binds COPII, Vps74/GOLPH3 (if its homolog exists in plants), or any other coat and adaptor proteins?

We appreciate that the identification of binding partners for ERD2 is an important point but it is part of a long term research project and outside the scope of this present paper.

Others are relatively minor points but should be carefully checked and corrected.

4. L67-70. This explanation is awful. I couldn't understand the meaning until I went back to the previous 2018 paper.

We completely agree and have addressed this in multiple sections of the text to make sure readers understand our rationale.

5. L120-121. Supplemental figure 3C. Where is it?

This was an oversight and has now been added to the supplementary figure.

6. L195-202. The figure referred to here should be Figure 5, not 6

We apologise for the mistake, in the meanwhile further data have been added and the figure 6 applies now.

7. L475-487. This part is awkward. It looks like a mixed-up version of copy and paste. In L487, what do black arrow-head and white arrow-head mean?

We have corrected the description and re-phrased, white arrow heads are now termed closed arrow heads and black arrow heads are now termed open arrow heads.

8. L768-770. This is awful again. From this explanation, nobody can understand the difference between B1, B2 and B3. I was totally at a loss at the first glance.

This figure has changed compared to the first submission and B1, B2 and B3 no longer appear. We double checked the new figure and the legends and have avoided any clarity issues.

9. L813. Supp Figure 3. Where is it?

This figure has also changed now and we have made the required corrections

10. L835-838. The explanation about FRAP is too poor in the legend. How did the authors choose ROIs? How long time were they bleached for? It is not good to force readers to look into Materials and Methods to get essential information.

The legend has been improved together with a better description in the material and methods.

11. L895-914. Again, this is a terribly unkind figure legend. What are m, A and AE!? I went back to the main text and Material and Methods section repeatedly but couldn't find the answer and gave up.

We agree that there is a lot to take in for the reader and we have spent time improving the legend

and the figure layout Basically Figure 7C shows the quantification of signals in figure 7A and further repeats.

12. L982-994. I am not sure Supplementary Figure 3 is relevant in this paper.

The purpose of the data set is to illustrate the quantitative nature of the transport assay which not many peers in the field will expect from plant protoplasts. If the editor feels that it goes too far we will gladly remove it.

Reviewer #3 (Remarks to the Author):

Denecke and co-workers studied the dynamics of the K/HDEL receptor in plant cells. They first showed that ERD2 fused with an additional N-terminal transmembrane domain and a YFP is able to complement the loss of endogenous ERD2 in both *Nicotiana benthamiana* and *P. patens*. Overexpression of 9 out of 12 different ERD2 from different sources reduced the secretion of amy fused with HDEL or KDEL. YFP-TM-HsERD2 localised exclusively in the Golgi complex, whereas addition of YFP to the C-terminal showed dual ER and Golgi localisation and was unable to retain amy-HDEL intracellularly.

Analysis of several C-terminal residues of ERD2 showed that K206 and K207 are dispensable for amy-HDEL retention, while leucines at positions -2 and -3 are required. The addition of small tags to the C-terminus of ERD2 also impaired the receptor's ability to retain amy-HDEL, except for HA tag. Note that retention of amy-HDEL correlates with the exclusive presence of ERD2 in the Golgi. The deletion of the two leucines did not affect ER exit, but most likely the retention of ERD2 in the Golgi.

Furthermore the deletion of the last 5 amino acid impaired the receptor ability to retain amy-HDEL. To examine the role of COPI/COPII on ERD2 localisation, they generated a chimera in which the C-terminus of ERD2 was replaced with that of p24. This chimeric receptor was retained in the ER and was unable to prevent amy-HDEL secretion unless the two terminal lysines were mutated. In addition, introduction of a Golgi retention signal into a non-functional TM-ERD2-YFP restored Golgi localization and partial retention activity of amy-HDEL.

Despite similar plasmid transfection levels, the expression of ERD2-HA was much lower than that of amy-HDEL, but still sufficient to reduce amy-HDEL secretion. Quantitative analysis indicated that 4.5 amy-HDEL molecules are retained in the cells per ERD2. Although, an accurate dose-response curve suggested that 2 molecules of ERD2 can retain up to 100 molecules of amy-HDEL.

In summary, this study using a number of artificial ERD2 constructs showed that perturbation of the K/HDEL receptor alters its intracellular distribution and ability to retain an overexpressed ligand (amy-HDEL). It also showed an association between the localisation of the receptor in the Golgi and its ability to retain amy-HDEL. While it is unclear what is happening when ERD2 has a mixed ER-Golgi distribution: in this case, although amy-HDEL promotes the redistribution of the receptor to the ER (Fig. 3D), it is not retained within the cells (Fig. 4C).

Experiments supporting a role of C-terminal leucines in interfering with COPI-dependent retrograde transport of the receptor are insufficient and rather indirect. For example, Co-IP experiments should be performed to understand how the ERD2 interactome changes by removing these leucines and thus possibly clarify the relationship with COPI. We believe that to date these experiments only suggest a role for leucines in slowing down retrograde receptor transport. The fact that introducing the C-terminal region of p24 to ERD2 alters the distribution and function of the chimeric proteins

may not be surprising. We cannot exclude an important effect on the three-dimensional organisation of the receptor.

In our opinion, the main problem is that the experiments do not support the message the manuscript wants to convey. The data presented do not demonstrate that endogenous ERD2 is involved in the recovery of endogenous K/HDEL proteins while remaining in the Golgi complex, and over 20 years of evidence suggests otherwise. We do not know which model is correct, but the evidence provided, based only on the overexpression of modified proteins, is not sufficient to change the common paradigm.

We disagree with the reviewer. The common paradigm is completely based on overexpressed modified proteins which can be equally described as artificial, as the reviewer has chosen to describe our constructs. Instead of indirect receptor-redistribution assays, we have chosen to develop a quantitative gain-of-function assay displaying the ability of native ERD2 to increase the retention of HDEL cargo. This is a direct assay, and it revealed that C-terminal fluorescent fusions of ERD2 have lost biological activity. A single inconsistency such as this should really be sufficient to re-evaluate a model. As the old model is very popular, our first paper raised a lot of concerns regarding the lack of genetic complementation, the plant-specific nature of our work and that we could not exclude that the ERD2 C-terminus has a fast ER export signal. These questions were fully addressed in our first submission and in addition we have described a whole series of further experiments, each of which had the potential to prove the old model. However, each of these experiments did the exact opposite.

As an example, the reviewer states that "it is unclear what is happening when ERD2 has a mixed ER-Golgi distribution". The truth is that we make it very clear what is happening; these ERD2 variants are no longer active and the dual localisation is an artefact, as is the ligand-induced redistribution. Yet we confirm that C-terminal fluorescent ERD2 fusions showed what proponents of the recycling model wanted to see, a dual ER-Golgi localisation. We should not have had to go through the trouble of repeating experiments with ERD2-YFP just so the field can see that we can also monitor ligand-induced redistribution. We did these experiments to accommodate peer pressure and protect our integrity.

We show using human ERD2 and plant ERD2 that a di-leucine motif mediates Golgi-retention and is required for activity. If instead a KKXXXXX sequence really plays a role in COPI-mediated recycling of receptor-ligand complexes, then our point mutations should have knocked out ERD2 activity completely. Instead, the mutations were silent. We do not wish to repeat all our other findings here, but in summary, our paper points at the fact that the last 30 years of evidence is flawed by the use of inactive ERD2 fusions and the reluctance of the field to develop methods measuring directly the transport of HDEL- and KDEL proteins quantitatively. To cite the fact that our results contradict the dogma as argument against publication is purely a circular argument and as such inappropriate.

Specific points:

The reviewer is concerned that the ERD2-P24 hybrid is perhaps misfolded, but appears to have missed that mutating the two lysines (KKSS) restores biological activity and also localises to the Golgi, so it is highly unlikely that the 9 amino acid swap changes the 3 dimensional organisation of the receptor. More dramatically, we could re-activate ERD2-YFP by adding an N-terminal TM domain and a Golgi-retention signal, so it appears that the ERD2 core structure is intact even when a large YFP is fused to the C-terminus (it simply masks the di-leucine motif).

The reviewer also argues that we should work directly with endogenous receptors and ligands, but no publications in the field have ever carried out such research. Everyone uses overexpressed protein fusions, whilst we have at least validated said fusions with activity assays, spotted a problem with ERD2-YFP and found a better way to fluorescently label ERD2 so that activity is maintained.

We agree that a model is difficult to prove, as a model can only fail to be disproven. Clearly, the recycling model was disproven multiple times and we encourage the field to accept this now.

.

Minor points

The manuscript is rather difficult to follow, most of the acronyms are undefined and the experiments are inadequately described, assuming that the reader is perfectly familiar with the previous manuscript by the same author (Fernanda et al The Plant Cell 2018), but this may not be the case for the wide readership of nature communication.

We agree that the manuscript was written in a rather condensed manner and this has now been improved.

REVIEWERS' COMMENTS

Reviewer #1 (Remarks to the Author):

In this revision, the authors have invested great effort with additional experiments to address my previous comments, albeit the underlying mechanism of ERD2 retention remains elusive. Wherever my comments disagree with the authors' conclusions, they made an effort to explain their point of view and I am happy with that. I strongly recommend publishing this paper, because it is relevant for the plant and mammalian cell biology community and will have a significant impact.

Best wishes,

Jennifer Schoberer

Reviewer #2 (Remarks to the Author):

I am satisfied by the revisions and the answers to reviewers' comments. I support the publication of the revised manuscript.

Reviewer #3 (Remarks to the Author):

The revised manuscript, by Denecke and co-workers, has been improved from the original version. The experiments are better described and any labelling errors or incorrect references to specific panels/figures have been corrected.

However, the main concerns we raised in the first phase of the revision remain.

First of all, I would like to make it clear that I completely trust the data presented in this study. The experiments are well thought out and the results are clear.

In my opinion, to go beyond previous conclusions obtained by eminent researchers and published in leading journals, the new evidence cannot be based on recombinant proteins alone. I do not exclude that the proposed model could be the right one, but at least it would have to be tested on

endogenous proteins in order to be published in a high-level journal. The authors decided not to respond to our simple request.

I disagree with the authors' arguments that no research has ever been conducted on endogenous KDELRs to demonstrate ligand-induced redistribution. Indeed there are several manuscripts reporting the retrotranslocation of endogenous KDELR following stimulation with different ligands (e.g. DOI: 10.1038/emboj.2012.134; DOI: 10.7554/eLife.68380).

Furthermore, one of the foundations of this manuscript, namely that the tagged KDELR receptor is non-functional, is disproved by important previous studies (e.g. Cell Report, DOI: 10.1016/j.celrep.2018.10.055).

However, I could bring an infinite amount of evidence to disprove the conclusions of this manuscript and I am sure that the authors can contradict all this evidence and explain the weakness of my arguments and the strength of their manuscript.

At this point, having emphasised my thoughts and expressed my concerns as to how much this data could add to the scientific community, I do not object to publication if the Editor feels the manuscript deserves to be published in a high impact journal.

This is also out of respect for my colleagues who have reviewed this manuscript and appreciated it.